# MicroRNA-27a controls the intracellular survival of *Mycobacterium tuberculosis* by regulating calcium-associated autophagy

Feng Liu[1], Jianxia Chen[1,2], Peng Wang[3,4], Haohao Li[3], Yilong Zhou[3], Haipeng Liu [1,2], Zhonghua Liu[1], Ruijuan Zheng[1], Lin Wang[3], Hua Yang[1], Zhenling Cui[1], Fei Wang[3], Xiaochen Huang[1], Jie Wang[1], Wei Sha[4], Heping Xiao[4] & Baoxue Ge[1,2,3]

Tuberculosis (TB) caused by *Mycobacterium tuberculosis* (*Mtb*) kills millions every year, and there is urgent need to develop novel anti-TB agents due to the fast-growing of drug-resistant TB. Although autophagy regulates the intracellular survival of *Mtb*, the role of calcium ($Ca^{2+}$) signaling in modulating autophagy during *Mtb* infection remains largely unknown. Here, we show that microRNA miR-27a is abundantly expressed in active TB patients, *Mtb*-infected mice and macrophages. The target of miR-27a is the ER-located $Ca^{2+}$ transporter CAC-NA2D3. Targeting of this transporter leads to the downregulation of $Ca^{2+}$ signaling, thus inhibiting autophagosome formation and promoting the intracellular survival of *Mtb*. Mice lacking of miR-27a and mice treated with an antagomir to miR-27a are more resistant to *Mtb* infection. Our findings reveal a strategy for *Mtb* to increase intracellular survival by manipulating the $Ca^{2+}$-associated autophagy, and may also support the development of host-directed anti-TB therapeutic approaches.

---

[1] Shanghai Key Lab of Tuberculosis, Shanghai Pulmonary Hospital, Tongji University School of Medicine, Shanghai 200433, China. [2] Clinical Translation Research Center, Shanghai Pulmonary Hospital, Tongji University School of Medicine, Shanghai 200433, China. [3] Department of Microbiology and Immunology, Tongji University School of Medicine, Shanghai 200092, China. [4] Department of TB, Shanghai Pulmonary Hospital, Tongji University School of Medicine, Shanghai 200433, China. These authors contributed equally: Feng Liu, Jianxia Chen, Peng Wang. Correspondence and requests for materials should be addressed to B.G. (email: baoxue_ge@tongji.edu.cn)

Mycobaterium tuberculosis (Mtb) remains the leading bacterial cause of death in humans throughout the world[1]. In 2015, Mtb infection was responsible for 10.4 million new cases and 1.4 million death[2]. However, only ~10% of infected individuals develop active tuberculosis (TB), while the remaining 90% of cases exhibit latent infection, indicating a critical role for host immunity in limiting Mtb infection[3].

Autophagy is a cellular process that can form double membrane-layered vesicles to deliver macromolecules, organelles, or intracellular pathogens for lysosomal degradation[4,5]. A number of molecules including autophagy-related ATG family proteins orchestrate signaling events that regulate autophagy flux including autophagosome initiation, elongation, maturation, and fusion with lysosomes[4–6]. As a secondary messenger, intracellular $Ca^{2+}$ acts as a major regulator of autophagy. Intracellular $Ca^{2+}$ is actively pumped out of the plasma membrane from the cytosol or is sequestered into multiple organelles including the endoplasmic reticulum (ER), mitochondria, and lysosomes[7]. Upon stimulation, entry of $Ca^{2+}$ into the cytoplasm activates CaMKKβ, a $Ca^{2+}$/calmodulin-dependent serine/threonine kinase[8]. Once activated, CaMKKβ phosphorylates AMPK on $Thr^{172}$, leading to its activation[8]. Activated AMPK has been shown to inhibit mTORC1, which phosphorylates the serine/threonine protein kinase ULK1 (unc-51-like autophagy activating kinase 1, the mammalian orthologue of yeast ATG1) on $Ser^{757}$, preventing the association of ULK1 with AMPK[9]. Phosphorylation of AMPK on $Thr^{172}$ by CaMKKβ thus enables AMPK to interact with and phosphorylate ULK1 on $Ser^{555}$. AMPK-phosphorylated ULK1 forms a complex with FIP200 (200 kDa focal adhesion kinase family-interacting protein) and ATG13 to induce autophagy by phosphorylating Beclin-1 and activating VPS34 lipid kinase[10]. Recently, it has been reported that lysosomal $Ca^{2+}$ released through mucolipin 1 (MCOLN1) activates the phosphatase calcineurin, which dephosphorylates TFEB, thus promoting its nuclear translocation to induce lysosomal biogenesis and autophagy[11]. However, whether and how non-lysosomal calcium signaling can also regulate autophagy still remains unclear.

Mtb, as a successful intracellular pathogen, that can survive and persist in macrophage by blocking the fusion of phagosome with lysosome[12]. Recent work demonstrated that autophagy played an important role in the containment of intracellular Mtb by targeting Mtb for lysosome degradation, bypassing Mtb-mediated blockade of phagosome maturation[13]. Stimulation of autophagy by autophagy-inducing agents or interferon-γ (IFN-γ) increased the delivery of Mtb to autophagosomes and lysosome-mediated bacterial killing[13–16]. However, in resting cells, the majority of Mtb is not found to be associated with an autophagosome component such as LC3. Moreover, when unstimulated, autophagy is only a minor contributor to the clearance of Mtb in macrophages and in vivo in mice[14,17–19], which may be due to the ability of Mtb to partly disable or manipulate autophagy through multiple mediators, such as secretory proteins ESAT-6/CFP-10 and PE_PGRS47[20,21], or the cell wall component lipoarbinomannan (LAM)[22].

MicroRNAs (miRNAs) are a family of small noncoding RNA molecules ~21–25 nucleotides long that bind to the 3′ untranslated region (3′ UTR) of target mRNAs and reduce protein expression by blocking mRNA translation and/or by promoting mRNA degradation[23,24]. Recent studies underscore the critical role of miRNAs in the evasion of Mtb from autophagic clearance of macrophages[25]. It has been shown that Mtb upregulates miR-33 to suppress several key autophagy effector molecules, such as ATG5, ATG12, LAMP1, LC3B, AMPK, and FOXO3 in macrophages to inhibit autophagy[26]. Kumar et al. have reported that Mtb downregulates the expression of miR-17 to enhance its target proteins Mcl-1, which in turn interacts with BECLIN-1 to inhibit

autophagy[27]. Upregulation of miRNA-125a in Mtb-infected macrophages targets UV radiation resistance-associated gene (UVRAG) to prevent autophagy[28]. More recently, Kim et al. found that miR-144* is induced by Mtb to target DNA damage regulated autophagy modulator 2 (DRAM2), resulting in suppression of autophagy[29]. However, our understanding of the role of miRNAs in modulating autophagic signaling pathways during Mtb infection and their clinical relevance is still limited.

In this study, we found that miR-27a is abundantly expressed in active TB patients, Mtb-infected animals and infected cells. Induction of miR-27a was found to directly target the $Ca^{2+}$ transporter Cacna2d3, and down-regulate ER $Ca^{2+}$ signaling to inhibit autophagy, thus promoting the intracellular survival of Mtb.

## Results

**Induction of miR-27a expression by Mtb.** We evaluated miRNA expression profiles from peripheral blood mononuclear cells (PBMCs) of the patients with active pulmonary TB, the lungs of Mtb strain H37Rv-infected C57/BL6 mice, and H37Rv-infected murine primary peritoneal macrophages. Using miRNA deep sequencing, we found that miR-27a, a microRNA mainly implicated in tumor cells[30–32], was commonly upregulated in all samples compared to their representative controls (Fig. 1a–c). As miR-27a's functional role in regulation of Mtb infection remains uncharacterized, we choose it for our further study. The upregulation of miR-27a was validated by quantitative real time PCR (RT-PCR) in PBMCs from human TB samples, the lungs of H37Rv-infected C57/BL6 mice, and H37Rv-infected macrophages (Supplementary Fig. 1a-c). However, Mtb-induced miR-27a expression was markedly attenuated in $Tlr2^{-/-}$ macrophages but not in $Tlr4^{-/-}$ macrophages (Supplementary Fig. 1d). Consistently, among the TB component examined, only stimulation with the TLR2 ligand peptidoglycan (PGN), but neither of the TLR4 ligand lipopolysaccharide (LPS) nor the Gram-negative bacterium Escherichia coli, TDM, ESAT-6 and ManLAM, induced the expression of miR-27a (Supplementary Fig. 1e). To further investigate the signaling pathway involved in the induction of miR-27a, we infected macrophages with Mtb in the presence of a selective p38 inhibitor (SB203580), MEK inhibitor (PD98059), or NF-κB inhibitor (pyrrolidine dithiolar-bamate, PDTC). Selective inhibition of the MEK-ERK pathway by PD98059 or NF-κB pathway by PDTC markedly reduced Mtb-induced miR-27a expression in macrophages (Supplementary Fig. 1f). These results suggested that Mtb infection may induce miR-27a expression via the activation of ERK MAPK and NF-κB pathways.

**miR-27a promotes intracellular survival of Mtb.** To examine the functional role of miR-27a during Mtb infection, we analyzed its effect on the intracellular survival of Mtb by colony-forming unit (CFU) assay. Primary murine peritoneal macrophages were transiently transfected with a miR-27a mimic or inhibitor and were then infected with H37Rv at an MOI of 5 for 24 h. As transfection of miR-27a mimic or inhibitor had no effect on phagocytosis of Mtb by macrophages, treatment with the miR-27a mimic significantly promoted the intracellular survival of Mtb in macrophages, whereas the miR-27a inhibitor reduced the intracellular survival of Mtb (Fig. 1d, e and Supplementary Fig. 2a). LDH assay were also performed and the result showed that miR-27a inhibitor or mimic did not affect the cytotoxicity of macrophages induced by Mtb infection (Supplementary Fig. 2b). To better characterize the function of miR-27a, we generated $miR-27a^{-/-}$ mice by using the CRISPR-Cas9 genome editing method[33] and qPCR results showed that miR-27a's expression but not the other members of the miR-23/24/27 cluster, was

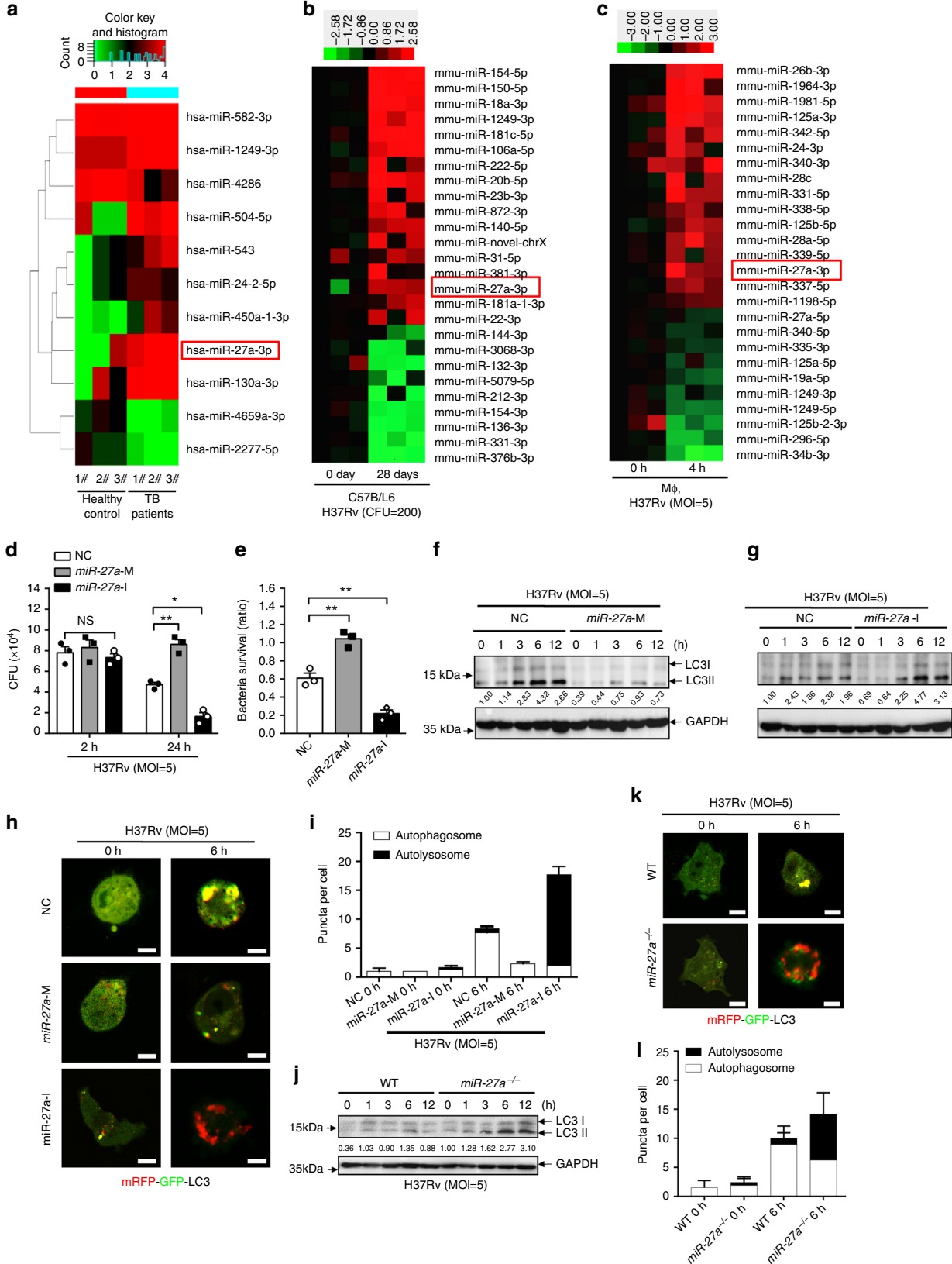

dramatically decreased in *miR-27a*[-/-] macrophages (Supplementary Fig. 2c and d). LDH assay showed that there was no cytotoxicity of *miR-27a*[-/-] macrophages induced by *Mtb* infection (Supplementary Fig. 2e) and CFU assay showed that genetic deletion of *miR-27a* in peritoneal macrophages also reduced the intracellular survival of *Mtb* (Supplementary Fig. 2f and g).

It has been reported that the intracellular survival of *Mtb* is controlled by autophagy[14]. To examine whether miR-27a enhances *Mtb* survival through autophagy, wild type (WT) or *miR-27a*[-/-] macrophages, or macrophages transfected with miR-27a mimic or negative control (NC) were first pretreated with either Bafilomycin A1 (Baf A1)[34], a vacuolar-ATPase

**Fig. 1** miR-27a promotes the intracellular survival of *Mtb* through autophagy. **a–c** 2D cluster analysis across miRNA probe (right) and subjects (bottom). Heatmap shows downregulated (green) and upregulated (red) miRNA from: (**a**) PBMCs of healthy donors (*n* = 3) and pulmonary TB patients (*n* = 3); (**b**) lung of the control or mice (*n* = 3) aerosol-infected with ~200 CFU *Mtb* H37Rv for 0 and 28 days; (**c**) murine primary peritoneal macrophages infected with H37Rv (MOI = 5) for indicated times. **d–i** macrophages pre-treated with NC, miR-27a-M (mimic), or miR-27a-I (inhibitor) were infected with *Mtb* for indicated times, and then subjected to (**d**, **e**) CFU detection; (**f**, **g**) Immunoblot (IB) of lysates; (**h**) confocal analysis of mRFP-GFP-LC3B spots, Bar, 5 μM. (**i**) Quantification of average number of LC3 puncta in each cell. (**j–l**) WT or *miR-27a*⁻/⁻ macrophages infected with *Mtb*, and then subjected to (**j**) IB of lysates; (**k**) confocal analysis of mRFP-GFP-LC3B spots; Bar, 5 μM. **l** Quantification of average number of LC3 puncta in each cell. (**d**, **e**, **i**, **l**) \*\**p* < 0.01 by the unpaired *t*-test. Data are from three independent experiments with biological duplicates in each (**d**, **e**, **i**, **l**; mean ± s.e.m. of *n* = 3 duplicates) or representative of three independent experiments (**f**, **g**, **h**, **j**, **k**)

inhibitor that suppresses the autophagic flux at the autophagosome–lysosome fusion stage, or AR-12[35], an autophagy inducer mainly targeting PDK-1, and were then infected with *Mtb*. The altered effect of miR-27a mimic or miR-27a deficiency on the intracellular survival of *Mtb* was not found in macrophages pretreated with Baf A1 or AR-12 (Supplementary Fig. 3a and b). Furthermore, WT and *Atg5*⁻/⁻ Raw 264.7 macrophages were transfected with miR-27a mimic or inhibitor, and then infected with *Mtb*. Similarly, the altered effect of miR-27a mimic or inhibitor on the intracellular survival of *Mtb* was no longer found in *Atg5*⁻/⁻ macrophages (Supplementary Fig. 3c and d). These results suggested that miR-27a may promote the intracellular survival of *Mtb* through modulating autophagy.

It has been shown that conversion of soluble LC3-I to lipid bound LC3-II is associated with the formation of autophagosomes, and such conversion is a key marker of autophagy[36]. Overexpression of miR-27a markedly reduced the amount of LC3-II, whereas the miR-27a inhibitor enhanced the amount of LC3-II (Fig. 1f, g). To further examine the functional role of miR-27a in autophagy flux during *Mtb* infection, we used the mRFP-GFP-LC3 reporter construct to monitor LC3 aggregation in autophagosomes and autophagolysosomes:[37] In pH-neutral autophagosomes, mRFP-GFP-LC3 produces a yellow signal, whereas in autophagolysosomes, a stronger red fluorescence signal is much more evident due to the loss of the pH-sensitive GFP signal. In macrophages infected with *Mtb*, miR-27a mimic markedly inhibited the formation of autophagosomes, whereas the miR-27a inhibitor accelerated autophagosome maturation as determined by the number of LC3 puncta and the extent of yellow-to-red conversion of the mRFP–GFP tandem-LC3 reporter[37] (Fig. 1h, i). Gene deletion of *miR-27a* in macrophages also markedly enhanced the production of LC3-II and the autophagosome–autophagolysosome conversion (Fig. 1j–l). Furthermore, WT and *miR-27a*⁻/⁻ macrophages were treated with chloroquine (CQ) and then infected with *Mtb* at MOI = 5, western blot results and mRFP-GFP-LC3 reporter assay showed that deficiency of *miR-27a* increased autophagosome formation and LC3-II amount in the presence of CQ (Supplementary Fig. 4a–c). Together, these data suggested that miR-27a may suppress the autophagic response to *Mtb* infection.

**miR-27a confers susceptibility to *Mtb* infection**. To assess the functional role of miR-27a in the pathogenesis of TB in vivo, WT mice or *miR-27a*⁻/⁻ mice were infected with the *Mtb* H37Rv strain for 28 days, and the bacterial burden and histopathological impairment in the lung of the infected mice were analyzed. Robust infiltration of neutrophil and lymphocyte were found in the lung tissue of WT mice, but *miR-27a*⁻/⁻ mice had less pathological impairment and infiltration of neutrophil and lymphocyte in their lungs (Fig. 2a, supplementary Fig. 5a–c). Moreover, compared to WT mice, *miR-27a*⁻/⁻ mice infected with *Mtb* exhibited a decreased bacterial burden in their lungs as indicated by both the CFU assay and acid-fast staining (Fig. 2b, c).

Manipulation of miRNAs to promote the host innate defense is a promising strategy in the development of novel therapeutic interventions against TB[38,39]. To investigate whether the beneficial effects of miR-27a suppression could be translated into a host-directed therapy (HDT), we used a miR-27a antagomir to inhibit miR-27a in a mouse *Mtb* infection model and tested its therapeutic efficacy. First, the mice were infected with *Mtb* via the aerosol route for 14 days, and were then intraperitoneally injected with a NC antagomir or miR-27a antagomir solution at 5 mg/kg for alternate days. At 28 days post-infection, qPCR results showed that miR-27a antagomir treatment dramatically suppressed miR-27a expression in lung tissues and spleens of the mice, and increased the expression of *Plk2* and *Pink1*, the two characterized miR-27a's targets, in the lung tissues (Supplementary Fig. 5d and e)[40,41]. Furthermore, the mice treated with miR-27a antagomir harbored a relatively lower bacterial load and displayed less pathological impairment in their lungs (Fig. 2d, e, Supplementary Fig. 5f & g). Together, these results demonstrated a therapeutic effect of miR-27a antagomir against *Mtb* infection in the murine model.

To further evaluate whether the therapeutic effects of miR-27a antagomir treatment are mediated through miR-27a, we infected the WT mice or *miR-27a*⁻/⁻ mice with *Mtb* H37Rv strain for 14 days, and then treated with NC antagomir or miR-27a antagomir for another 14 days. Treatment with the miR-27a antagomir lead to a reduction in bacterial load and less pathological impairment in the lungs of WT mice, but not in *miR-27a*⁻/⁻ mice (Figs. 2d, e, supplementary Fig. 5h). Similarly, the decreased bacterial burden caused by the miR-27a antagomir in the lung of WT mice was not observed in the *miR-27a*⁻/⁻ mice infected with *Mtb* (Fig. 2f), suggesting that the anti-TB therapeutic effect of the miR-27a antagomir may function through inhibition of miR-27a.

**miR-27a directly targets *Cacna2d3***. Using base alignment approach, a series of transcripts were found as potential targets of miR-27a, including *Plk2*, *Ror1*, *St6galnc3*, *Trim23*, *Akirin* and *Cacna2d3*(*Cac3*), and we chose *Cac3* for the next-step study. *Cacna2d3*, which encode a subunit of the voltage-dependent calcium (Caᵥ) channel complex[42,43], was found to have a specific binding site for miR-27a in its 3′-untranslated region (3′-UTR) (Fig. 3a). To confirm whether *Cacna2d3* is a direct target of miR-27a, we generated luciferase reporter constructs by cloning either the wildtype or a seed sequence-mutated (mut) 3′-UTR of *Cacna2d3* into the 3′-UTR of a pGL-3M-Luc vector. The mutated 3′-UTR of *Cacna2d3* had four bases changed (GUACUGUA to GGACGGGU), at the putative miR-27a-binding sites (Fig. 3a). We co-transfected these vectors together with the miR-27a mimic into HEK293T cells and analyzed the lysates 48 h later. Transfection with the miR-27a mimic markedly inhibited the luciferase activity for the wild-type 3′-UTR of *Cacna2d3*, but showed no significant repressive effect on the mutated 3′-UTR of *Cacna2d3* when compared to the control dsRNA (Fig. 3b). These results

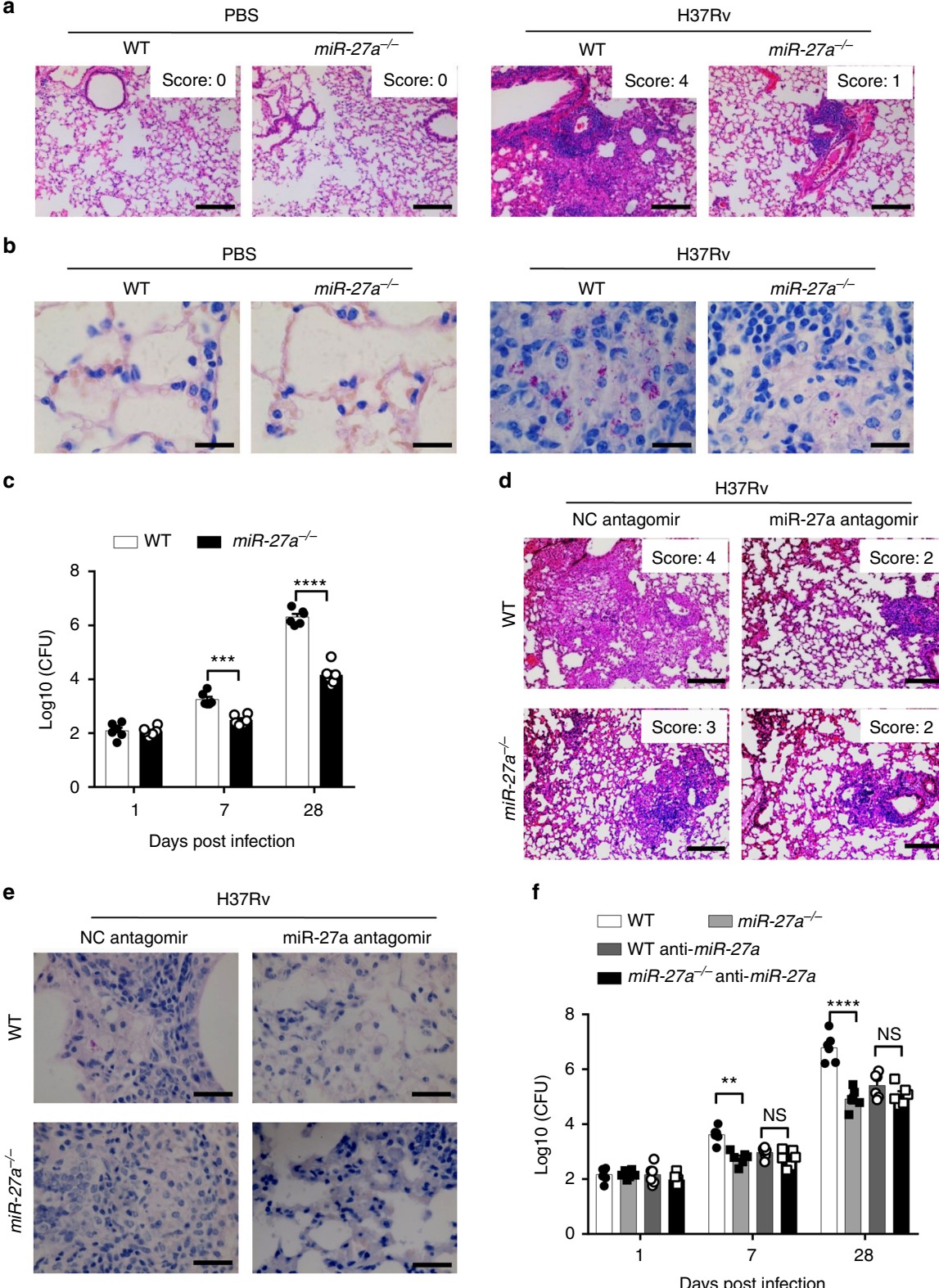

**Fig. 2** miR-27a controls susceptibility to *Mtb*. **a**–**c** WT or *miR-27a⁻/⁻* mice were infected with *Mtb* H37Rv for 28 days, and the lungs were subjected to (**a**) H&E staining. Scale bar, 100 μm; (**b**) acid-fast staining of bacteria. Scale bar, 20 μm. (**c**) CFU assay. **d**–**f** Mice were infected with *Mtb* for 28 days, and treated with NC antagomir or miR-27a antagomir alternate days, and the lungs were subjected to (**d**) H&E staining, Scale bar, 100 μm; (**e**) Acid-fast staining of bacteria. Scale bar, 20 μm. (**f**) CFU assay. *$p < 0.05$ and **$p < 0.01$ by the Mann-Whitney *U* test (**c**, **f**). Data are representatives of three independent experiments with biological duplicates in each (**c**, **f**; mean ± s.e.m. of $n = 6$ duplicates) or representative of three independent experiments (**a**, **b**, **d**, **e**)

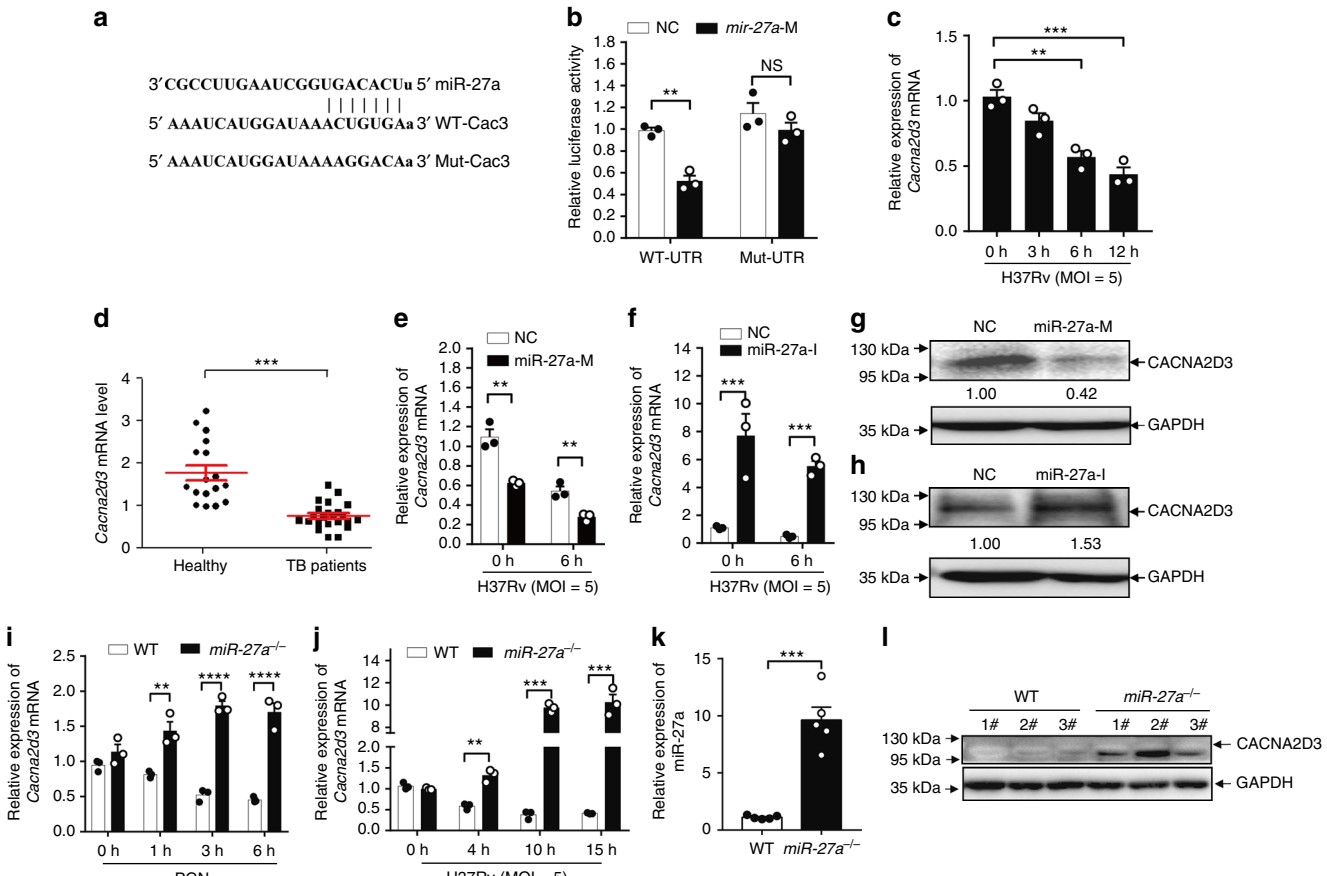

**Fig. 3** miR-27a directly targets Cacna2d3. **a** Predicted binding sites of miR-27a and *Cacna2d3* 3'-UTR. **b** Luciferase activity of HEK293T cells transfected with either WT or the seed sequence-mutated (mut) 3'-UTR of *Cacna2d3* plus scramble and miR-27a mimic. **c-f** Real-time PCR of the *Cacna2d3* mRNA in: (**c**) macrophages infected with *Mtb*; (**d**) PBMC from healthy donors and TB patients; (**e**, **f**) macrophages pre-treated with miR-27a-M (**e**) or-I (**f**), and then infected with *Mtb*. **g**, **h** IB of the lysates from macrophages pre-treated with miR-27a-M or miR-27a–I, and then infected with *Mtb*. **i** Real-time PCR of *Cacna2d3* in WT or *miR-27a*^-/- macrophages infected with *Mtb*. **j** Real-time PCR of *Cacna2d3* in WT and *miR-27a*^-/- macrophages treated with PGN for indicated times. **k** Real-time PCR of *Cacna2d3* in lung tissues from WT and *miR-27a*^-/- mice. **l** IB of lysates from WT and *miR-27a*^-/- macrophages ($n = 3$). NS, not significant ($p > 0.05$), *$p < 0.05$, **$p < 0.01$, ***$p < 0.001$, and ****$p < 0.0001$ by the unpaired *t*-test (**b-f**, **i**, **j**, **k**). Data are from three independent experiments with biological duplicates in each (**b-f**, **i**, **j**, **k**; mean ± s.e.m. of $n = 3$ duplicates) or representative of three independent experiments (**g**, **h**, **l**)

suggest that miR-27a may suppress the expression of *Cacna2d3* by binding to the 3'-UTR of *Cacna2d3* in a direct and sequence-specific manner.

The mRNA level of *Cacna2d3* was much lower in murine primary macrophages infected with *Mtb*, and in PBMCs from active TB patients (Fig. 3c, d) compared to their representative controls. Pearson's correlation analysis revealed that the expression of miR-27a was negatively correlated with the mRNA level of *Cacna2d3* in TB patients (Supplementary Fig. 6). To examine the effect of miR-27a on the expression of *Cacna2d3* during *Mtb* infection, murine primary macrophages were transfected with the miR-27a mimic or inhibitor and were then infected with *Mtb*. Treatment with the miR-27a mimic significantly inhibited the mRNA and protein levels of *Cacna2d3*, whereas the miR-27a inhibitor markedly enhanced both the mRNA and protein level of *Cacna2d3* in *Mtb*-infected macrophages (Fig. 3e–h). Similarly, genetic deletion of miR-27a increased the mRNA level of *Cacna2d3* in PGN-treated macrophages (Fig. 3i), as well as the mRNA and protein levels of *Cacna2d3* in *Mtb*-infected macrophages, and lungs of mice infected with *Mtb* as compared to their representative controls (Fig. 3j–l). These results suggested that induction of miR-27a suppress the expression of *Cacna2d3* during *Mtb* infection. Furthermore, of the other candidate genes of miR-27a, only expression of *Plk2* and *St6galnac3* mRNAs were

slightly increased in *Mtb*-infected *miR-27a*^-/- macrophages, when compared to WT macrophages (Supplementary Fig. 7a–e), suggesting that the induction of miR-27a may also suppressed expression of *Plk2* and *St6galnac3* during *Mtb* infection.

**Cacna2d3 inhibits intracellular survival of Mtb.** To dissect the functional role of CACNA2D3 in the regulation of *Mtb* infection, we performed a CFU assay to analyze its effects on the intracellular survival of *Mtb* in macrophages. Transfection of macrophages with specific siRNAs for *Cacna2d3* significantly increased *Mtb* viability (Fig. 4a, Supplementary Fig. 8a & 8c). Furthermore, we introduced *Cacna2d3* mutant mice which contain a piggyback (PB) insertion in the intron region between exon 27 and exon 28 and disrupted *Cacna2d3* expression (Supplementary Fig. 8b) [44–46]. Similarly, Suppression of *Cacna2d3* by PB insertion (*Cac3*^PB/PB) also increased the intracellular survival of *Mtb* in primary peritoneal macrophages (Fig. 4b, Supplementary Fig. 8d). Transmission electron microscopy (TEM) analysis revealed that relatively more bacteria are accumulated in *Cac3*^PB/PB macrophages compared to WT cells (Fig. 4c, Supplementary Fig. 8e). *Cac3*^PB/PB macrophages also showed a reduced amount of LC3-II when infected with *Mtb* (Fig. 4d). Confocal microscopy analysis showed that inhibition of *Cacna2d3* either by siRNA or PB

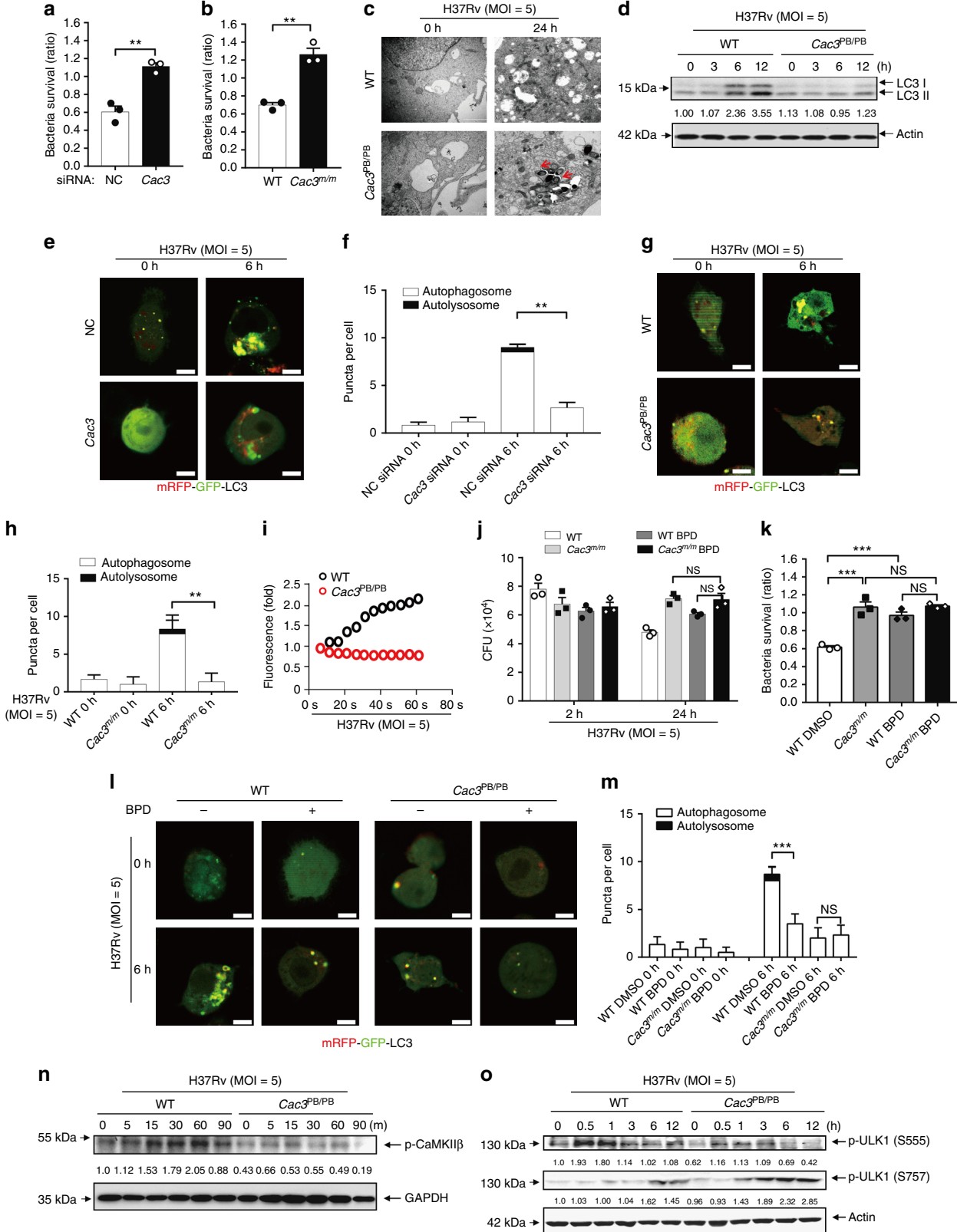

suppression significantly reduced the accumulation of LC3 puncta and autophagy flux (Fig. 4e–h). Furthermore, WT and *Cac3*<sup>PB/PB</sup> macrophages were treated with CQ and then infected with *Mtb*, western blot results showed that PB suppression of *Cac3* also decreased LC3-II amount in the presence of CQ (Supplementary Fig. 8f). These results suggested that *Cac3* may promote autophagy and decrease the intracellular survival of *Mtb*.

*Mtb* infection markedly increased the intracellular concentration of $Ca^{2+}$ flux. Treatment of $Ca^{2+}$ channel blockers nilvadipine[47] (Supplementary Fig. 9a) or amlodipine[48] (Supplementary Fig. 9d) markedly reduced the intracellular concentration of $Ca^{2+}$ in *Mtb*-infected macrophages. Inhibition of $Ca^{2+}$ also reduced the amount of LC3-II (Supplementary Fig. 9b & 9e) and the accumulation of LC3 puncta (Supplementary Fig. 9c

**Fig. 4** Cacna2d3 regulates survival of *Mtb* via $Ca^{2+}$-mediated autophagy. **a, b** Bacterial survival of *Mtb* in (**a**) primary macrophages treated with *Cacna2d3* siRNA; (**b**) macrophages from WT and *Cacna2d3* mutant ($Cac3^{PB/PB}$) mice, after infection with *Mtb* (MOI = 5) for 24 h, compared to 2 h. **c, d** WT or $Cac3^{PB/PB}$ macrophages were infected with *Mtb*, and then subjected to (**c**) TEM analysis; red arrow indicated the bacteria. (**d**) IB of lysates. **e–h** Confocal detection of mRFP-GFP-LC3B spots in *Cacna2d3*-siRNA-treated macrophages (**e, f**) or $Cac3^{PB/PB}$ macrophages, after infection with *Mtb* (MOI = 5) for 6 h (**g, h**); Bar, 5 μM. **i** Calcium assay in WT or $Cac3^{PB/PB}$ macrophages infected with *Mtb* (MOI = 5) for the duration of 1 min. **j, k** CFU assay (**j**) and intracellular survival of *Mtb* (**k**) in WT or $Cac3^{PB/PB}$ macrophages pretreated with Bepridil (BPD) (10 μM), and then infected with *Mtb* (MOI = 5) for 2 h or 24 h. **l, m** Confocal detection of mRFP-GFP-LC3B spots in WT or $Cac3^{PB/PB}$ macrophages pretreated with DMSO or BPD infected with *Mtb* at MOI 5 for 6 h. Bar, 5 μM. **n, o** IB of lysates for phosphorylated-CaMKKβ (**n**) and phosphorylated-ULK1 (**o**) from WT or $Cac3^{PB/PB}$ macrophages infected with *Mtb* for indicated times. NS, not significant ($p > 0.05$), * $p < 0.05$ and ** $p < 0.01$ by the unpaired *t*-test (**f, h**) or Mann-Whitney *U* test (**a, b, j**). Data are from three independent experiments with biological duplicates in each (**a, b, f, h, j, k, m**; mean ± s.e.m. of $n = 3$ duplicates) or representative of three independent experiments (**c–e, g, i, l, n, o**)

& 9f), indicating an essential role of $Ca^{2+}$ signaling in the regulation of autophagy during *Mtb* infection. CACNA2D3 is a subunit of the $Ca_v$ channel that drives $Ca^{2+}$ influx to activate synaptic neurotransmitter release. Suppression of *Cacna2d3* by PB insertion markedly reduced the intracellular concentration of $Ca^{2+}$ in *Mtb*-infected macrophages (Fig. 4i). To investigate whether CACNA2D3 regulated the intracellular survival of *Mtb* through $Ca^{2+}$ flux, we treated the WT or $Cac3^{PB/PB}$ macrophages with the a membrane-permeable calcium channel blocker, Bepridil (BPD) [49], which could block the cytosolic calcium release, and then infected those cells with the *Mtb* H37Rv strain for 24 h. The inhibitory effect of CACNA2D3 on the *Mtb* viability was not found in BPD-treated $Cac3^{PB/PB}$ macrophages (Fig. 4j, k). Similarly, treatment of BPD did not reduce the accumulation of LC3 puncta in the $Cac3^{PB/PB}$ macrophages as it did in WT macrophages (Fig. 4l, m), suggesting that CACNA2D3 may regulate the intracellular survival of *Mtb* via dampening $Ca^{2+}$ influx-mediated autophagy.

Regulation of autophagy by $Ca^{2+}$ is dependent on the activation of CaMKK, a $Ca^{2+}$/calmodulin-dependent serine/threonine kinase[8]. The phosphorylation level of CaMKKβ at $Thr^{286}$ was dramatically decreased in *Mtb*-infected $Cac3^{PB/PB}$ macrophages compared to WT macrophages (Fig. 4n). When phosphorylated, CaMKKβ activates AMPK, which then phosphorylates ULK1 at $Ser^{555}$ to initiate autophagy[9]. On the other hand, mTOR mediates the phosphorylation of ULK1 at $Ser^{757}$ to inhibit autophagy by disassociating AMPK-ULK1[9]. Suppression of *Cacna2d3* markedly reduced the phosphorylation of ULK1 on $Ser^{555}$, but resulted in increased phosphorylation level on $Ser^{757}$ in *Mtb*-infected macrophages (Fig. 4o). It has also been shown that $Ca^{2+}$ signaling controls the activities of the phosphatase Calcineurin and of its substrate TFEB, a master transcriptional regulator of lysosomal biogenesis and autophagy[11]. *Mtb* infection markedly increased the nuclear translocation of TFEB in macrophages, but the neither the *Cacna2d3* suppression nor the depletion of *miR-27a* had significant effect on the activation of TFEB during *Mtb* infection (Supplementary Fig. 10). Together, these results suggest that CACNA2D3 may regulate autophagy mainly through $Ca^{2+}$-induced activation of CaMKKβ/ULK1.

Intracellular $Ca^{2+}$ could be released from multiple organelles, including the ER, mitochondria, and lysosomes[7]. Immunostaining analysis showed co-localization of CACNA2D3 with the ER marker Calnexin, but not with the lysosome marker LAMP1 in *Mtb*-infected macrophages, suggesting that CACNA2D3 is mainly located on ER (Supplementary Fig. 11a, b). We therefore used the ER calcium chelator *N,N,N',N'*-Tetrakis (2-pyridylmethyl) ethylenediamine (TPEN) to examine its effect on autophagy flux during *Mtb* infection[50]. Treatment with TPEN significantly reduced the number of LC3 puncta and autophagolysosome formation (Supplementary Fig. 11c, d), suggesting an essential role for ER $Ca^{2+}$ in *Mtb*-induced autophagy.

**CACNA2D3 promotes resistance to *Mtb* infection**. To investigate the functional importance of CACNA2D3 in the regulation of *Mtb* infection in vivo, WT or $Cac3^{PB/PB}$ mice were infected with H37Rv at CFU = 200 for 28 days. The lung tissues of $Cac3^{PB/PB}$ mice showed more severe histology impairment and more lymphocytes or neutrophil recruitment as compared to WT mice (Fig. 5a, Supplementary Fig. 12). Furthermore, the bacterial titers in the lung tissues of $Cac3^{PB/PB}$ mice were much higher than those of WT mice (Fig. 5b, c). Lastly, immunostaining of the lung tissues of $Cac3^{PB/PB}$ mice showed less LC3 protein compared to WT mice (Fig. 5d). These results suggested that CACNA2D3 may protect mice from *Mtb* infection by promoting autophagy.

**miR-27a regulates *Mtb* infection via *Cacna2d3* and $Ca^{2+}$**. To determine whether miR-27a regulates the intracellular survival of *Mtb* via targeting of *Cacna2d3*, WT or $Cac3^{PB/PB}$ macrophages were transfected with miR-27a mimic or inhibitor and were then infected with *Mtb* for 24 h. CFU assay and confocal microscopy analysis were performed to detect the intracellular bacterial load and the autophagic responses, respectively. Transfection with the miR-27a mimic significantly increased the bacterial survival of *Mtb* in WT macrophages, but not in $Cac3^{PB/PB}$ macrophages (Fig. 6a, Supplementary Fig. 13a). In contrast, the miR-27a inhibitor significantly reduced the bacterial survival of *Mtb* in WT macrophages, but not in $Cac3^{PB/PB}$ macrophages (Fig. 6b, Supplementary Fig. 13b). Confocal microscopy analysis showed that the formation of LC3 puncta in WT macrophages is inhibited by the miR-27a mimic, but is enhanced by the miR-27a inhibitor. However, no significant effect due to the miR-27a mimic or inhibitor on the formation of LC3 puncta in $Cac3^{PB/PB}$ macrophages was observed (Fig. 6c, d). These results suggested that miR-27a may inhibit autophagy to promote the intracellular survival of *Mtb* by targeting *Cacna2d3*. Furthermore, RNAi of *Plk2* or *St6galnac3*, the other two candidate targets of *miR-27a*, also increased the intracellular survival of *Mtb* in *miR-27a*-deficient macrophages (Supplementary Fig. 13c). However, RNAi of *Pink1*, another characterized *miR-27a* target in regulation of mitophagy[41], decreased intracellular survival of *Mtb* (Supplementary Fig. 13d & e), suggesting that *Pink1*, may not be the downstream target of miR-27a in regulation of *Mtb* survival. Together, all these results indicated that miR-27a may simultaneously target multiple transcripts, resulting in potent cumulative effects on *Mtb* viability.

Xenophagy is a selective autophagy that delivers the *Mtb* into lysosome for degradation. We therefore examined whether *miR-27a* inhibition or PB suppression of *Cacna2d3* affect the co-localization of *Mtb* H37Rv with p62, which tags bacteria for sequestration within LC3-positive autophagosomes. Macrophages transfected with miR-27a inhibitor contained more p62-positive *Mtb*, when compared to those macrophages transfected with NC. However, the increased p62-*Mtb* co-localization by miR-27a

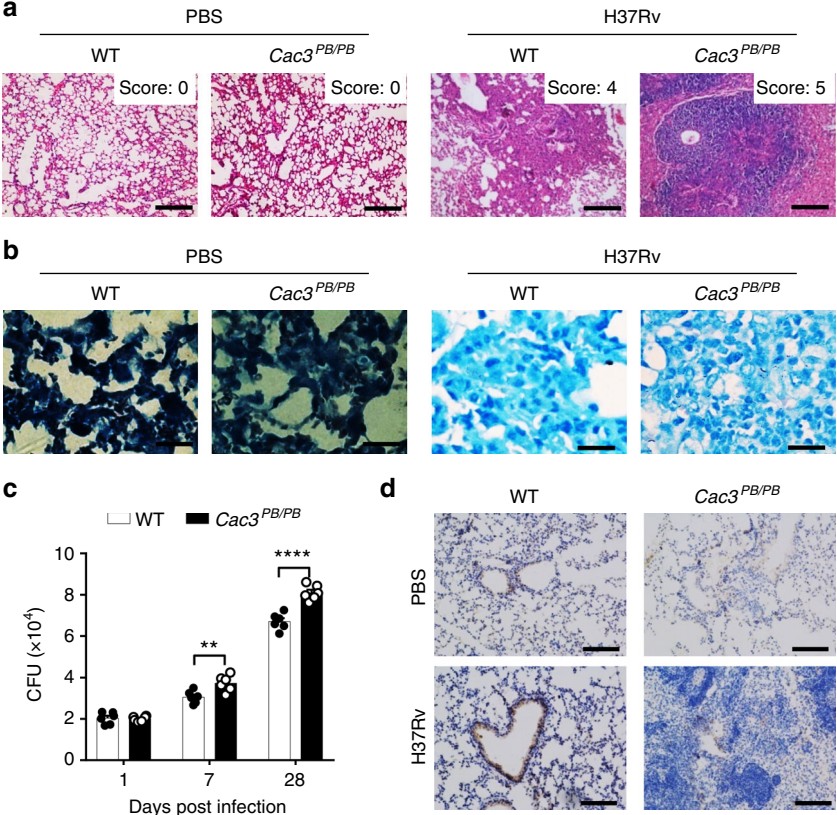

**Fig. 5** *Cacna2d3*-mutant mice are more susceptible to *Mtb* infection. FVB and *Cac3*[PB/PB] mice were infected with *Mtb* or treated with PBS for 28 days, and the lungs were subjected to (**a**) H&E staining. Scale bar, 100 µm. **b** Acid-fast staining of bacteria. Scale bar, 20 µm. **c** CFU assay. **d** Immunohistochemistry of LC3B. Brown signal indicated LC3. Scale bar, 100 µm. \*\**p* < 0.01, \*\*\*\**p* < 0.0001 by the Mann–Whitney *U* test (**c**). Data are representatives of three independent experiments. In **c**, data are representatives of three independent experiments with biological duplicates in each (**c** mean ± s.e.m. of *n* = 6 duplicates)

inhibition was totally attenuated in *Cac3*[PB/PB] macrophages (Supplementary Fig. 14a, c). Likewise, macrophages transfected with miR-27a inhibitor contains more LC3-positive *Mtb* when compared to macrophages transfected with NC, while *Cac3*[PB/PB] macrophages contains less LC3-positive *Mtb*, even pre-treated with miR-27a inhibitor (Supplementary Fig. 14b, d). These results suggested that miR-27a may also suppress the xenophagy of *Mtb* clearance via inhibiting *Cac3*.

We next examined whether miR-27a regulates the intracellular survival of *Mtb* through $Ca^{2+}$ signaling. Transfection with the miR-27a mimic markedly inhibited the *Mtb*-induced $Ca^{2+}$ influx, whereas the miR-27a inhibitor enhanced the $Ca^{2+}$ influx (Fig. 6e). Similarly, in responses to *Mtb* infection, *miR-27a*[-/-] macrophages showed enhanced $Ca^{2+}$ influx compared to WT macrophages (Fig. 6f). Pretreatment with the $Ca^{2+}$ inhibitor BPD eliminated the reduced survival of *Mtb* observed in *miR-27a*[-/-] macrophages compared to WT macrophages (Fig. 6g, h). The enhanced formation of LC3 puncta was also not found in *miR-27a*[-/-] macrophages pretreated with BPD (Fig. 6i, j). These results suggested that miR-27a may promote the survival the *Mtb* through downregulation of $Ca^{2+}$ signaling.

To examine the relevance of miR-27a with *Cacna2d3* in vivo, WT or *Cac3*[PB/PB] mice were infected with *Mtb* for 14 days, and were then treated with the NC antagomir or miR-27a antagomir every 3 days for another 14 days. The lungs of infected mice were then subjected to H&E staining. Treatment with the miR-27a antagomir alleviated the pathological impairments (Fig. 6k). Acid-fast staining and the CFU assay showed significantly lower bacterial titers in the lung of *Mtb*-infected WT mice, but not

those of *Mtb*-infected *Cac3*[PB/PB] mice (Fig. 6l, m), suggesting that miR-27a may regulate *Mtb* infection by targeting *Cacna2d3* in vivo.

## Discussion

*Mtb* is an intracellular pathogen that can evade lysosomal degradation and establish persistent infections. Although several studies have shown that *Mtb* promotes intracellular survival by inhibiting autophagy through secretory proteins or the induction of miRNAs, but how *Mtb* manipulates $Ca^{2+}$ signaling to regulate autophagy and promote its intracellular survival remains unknown. Here, we demonstrated that *Mtb* infection induces the expression of miR-27a in *Mtb*-infected cells, animals and patients. Induction of miR-27a direct targets and reduces the expression of the $Ca^{2+}$ transporter subunit CACNA2D3, thus, downregulating ER $Ca^{2+}$ signaling to inhibit autophagy. These data not only indicated miR-27a and CACNA2D3 as regulators of host responses against *Mtb* infection, but also revealed a strategy for *Mtb* to manipulate the ER $Ca^{2+}$ signaling to inhibit the host autophagy (Supplementary Fig. 15). Furthermore, treatment with a miR-27a antagomir significantly ameliorates the pathogenesis of *Mtb* infection, suggesting an intervention strategy for TB.

Increasing studies have shown that miRNAs are involved in host responses to *Mtb* infection[25,51]. Our data demonstrated that miR-27a is not only highly expressed in *Mtb*-infected cells and animals, but is also much higher in active TB patients, suggesting that there is a clinical relevance of this microRNA with the pathogenesis of TB. Induction of miR-27a inhibited the

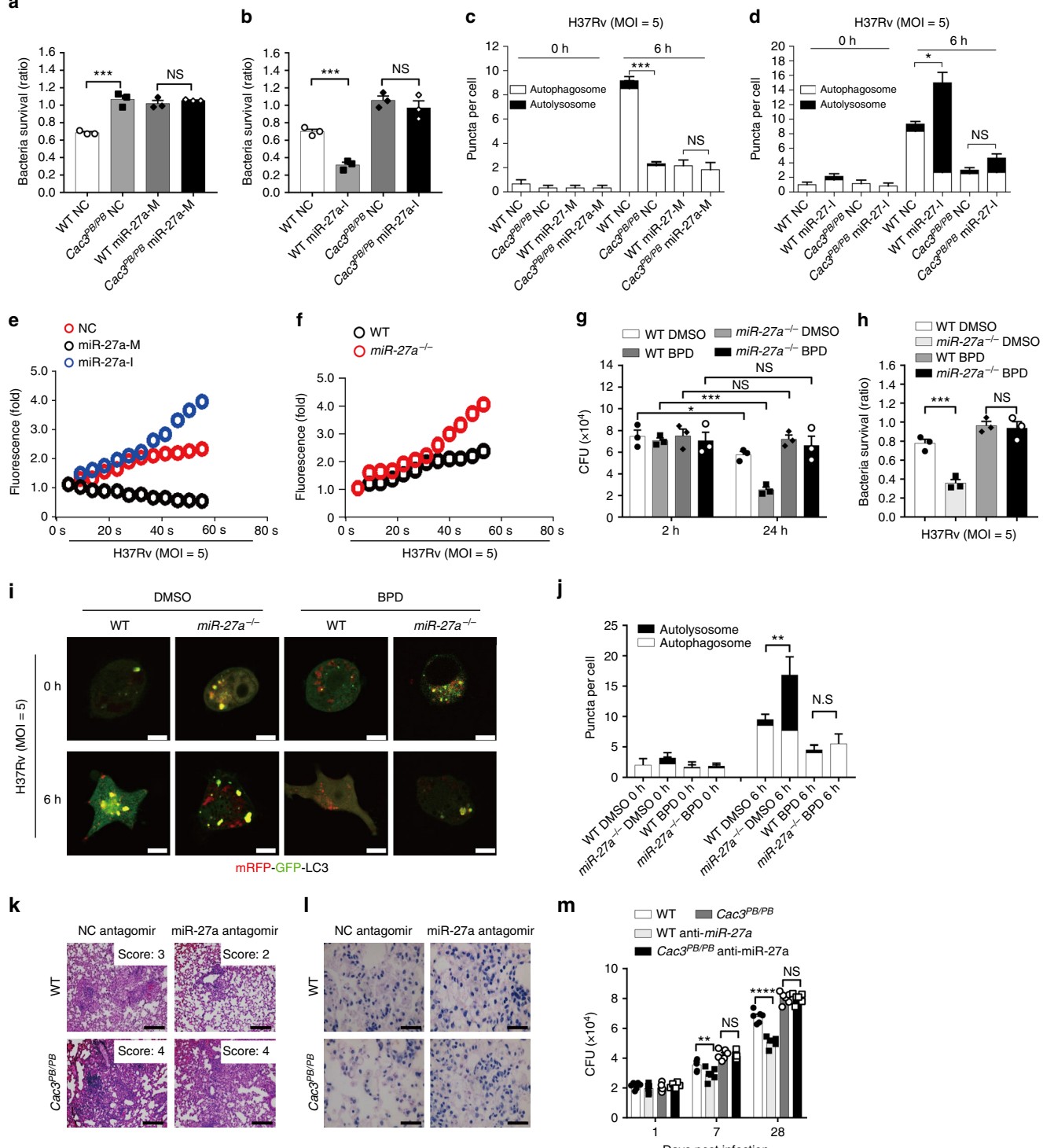

**Fig. 6** miR-27a regulates *Mtb* infection via Cacna2d3/Ca$^{2+}$. **a**, **b** Intracellular survival of WT or *Cac3$^{PB/PB}$* macrophages pretreated with (**a**) NC or miR-27a-M, (**b**) NC or miR-27a-I, and then infected with *Mtb* for 2 h and 24 h. **c**, **d** LC3 puncta from macrophages pretreated with NC and miR-27a-M (**c**) or miR-27a-I (**d**) then infected with *Mtb*. **e**, **f** Intracellular calcium fluorometry assay of (**e**) NC, miR-27a-M or miR-27a-I-treated, or (**f**) WT or *miR-27a$^{-/-}$* macrophages, and then infected with *Mtb* for 1 min. **g**, **h** CFU assay (**g**) and intracellular survival of *Mtb* (**h**) of macrophages pretreated with DMSO or Bepridin (BPD), and then infected with *Mtb* for 2 and 24 h. **i**, **j** Assay of (**i**) mRFP-GFP-LC3B spots or (**j**) numbers of LC3 puncta in macrophages of WT and *miR-27a$^{-/-}$* mice pretreated with DMSO or BPD, and then infected with *Mtb*. Bar, 5 μM. **k**, **l** FVB and *Cac3$^{PB/PB}$* mice treated with NC antagomir or miR-27a antagomir twice a week for 1 month after *Mtb* infection for 15 days, and then subjected to (**k**) H&E staining of mice lung, Scale bar, 100 μm; (**l**) acid-fast staining of bacteria in the mice lung, Scale bar, 20 μm; **m** CFU in mice lung. NS, not significant ($p > 0.05$), * $p < 0.05$ and ** $p < 0.01$ by the unpaired *t*-test (**c**, **d**, **h**, **j**) or Mann–Whitney *U*- test (**a**, **b**, **g**, **m**). Data are representatives from three independent experiments with biological duplicates in each (**a**, **b**, **g**, **h**, mean ± s.e.m. of *n* = 3 duplicates; **c**, **d**, **j**, **m**, mean ± s.e.m. of *n* = 6 duplicates) or representatives of three independent experiments (**e**, **f**, **i**, **k**, **l**)

autophagy to promote the intracellular survival of *Mtb* in macrophages. Thus, the induction of miR-27a to inhibit the autophagy appears to be a mechanism that could be used by *Mtb* as a strategy to evade the host immune response. Furthermore, treatment with a miRNA-27a antagomir protected mice from *Mtb* infection as indicated by reduced pathological impairment in the lung and a lower bacterial load, suggesting that miRNA-27a inhibitors could be repurposed for host-directed therapy of TB.

Though our data showed that TLR2 is essential for the induction of miRNA-27a during *Mtb* infection, but TLR2 only confers marginal protection against mycobacterial infection[52]. It is possible that TLR2 activates multiple branches of downstream signaling and stimulates both anti-bacterial effectors and pro-bacterial factors. The possible antagonism of these two arms of responses underscore the importance to specifically target factors involved in the immune evasion strategies employed by *Mtb*. Other studies have reported that miR-27a targets various proteins that participate in different cellular functions including tumorigenesis, hypoxia, and ischemia[31,32]. Moreover, whether miR-27a may exert its function in the pathogenesis of TB through targeting other proteins warrants further clarification.

We found that miR-27a directly targets *Cacna2d3*, a α2δ3 subunit of the Cav channel. *Cacna2d3* was previously implicated to be involved in several tumors, including lung cancer, breast cancer, gastric cancer, and neuroblastoma[53], however, a comprehensive study on the mechanism and extent its tumor suppressive function is still lacking. Our data demonstrated that *Mtb* downregulated the expression of *Cacna2d3*. Genetic inhibition of *Cacna2d3* almost totally abrogated the intracellular $Ca^{2+}$ influx, suggesting that the α2δ3-containing Cav channel as the main entry pathway for $Ca^{2+}$ influx induced by *Mtb* infection. Interestingly, PB suppression of *Cacna2d3* impaired autophagosome formation and was accompanied with higher intracellular survival of *Mtb*. Furthermore, *Cacna2d3*-deficient mice had much higher bacterial titers and were more susceptible to *Mtb* infection. Thus, our findings reveal a function for *Cacna2d3* as an essential host autophagic factor for the *Mtb* control. Whether the autophagic effect of *Cacna2d3* is associated with its anti-tumor function remain to be explored.

$Ca^{2+}$ is known to play a crucial role in TB pathogenesis. The ability of *Mtb* to manipulate $Ca^{2+}$ signaling to prevent phagosome maturation and phagosome–lysosome fusion has been extensively studied[54–56], but the exact role of $Ca^{2+}$ signaling in the regulation of autophagy during *Mtb* infection remains unknown. We found that a $Ca^{2+}$ blockade by specific inhibitors markedly reduced autophagosome formation and was accompanied by a higher bacterial load of *Mtb*, suggesting an essential role of $Ca^{2+}$ in autophagy and intracellular survival of *Mtb*. Intracellular $Ca^{2+}$ is actively released to the cytosol from the ER, mitochondria, and lysosomes, and a more recent study revealed that lysosomal $Ca^{2+}$ release through MCOLN1 activates TFEB to regulate autophagy[11]. We found that CACNA2D3 is mainly located on the ER membrane, and that a blockade of ER $Ca^{2+}$ signaling markedly inhibits *Mtb*-induced autophagosome formation, suggesting a mechanism by which ER $Ca^{2+}$ signaling is linked to autophagy during *Mtb* infection. Of note, though being a regulatory α2δ subunit of voltage gated calcium channels required for the assembly of the channel on the membrane, the increase in the cellular abundance of CACNA2D3 only showed modest (~2 fold) increasing effect on calcium signaling, most likely due to the loose association of α2δ subunit with the channel[57,58] and a possible saturation of channel assembly on the membrane.

Activation of autophagic processes by $Ca^{2+}$ signaling is dependent on the activation of CaMKK or TFEB. Our data demonstrated that *Cacna2d3*-mediated $Ca^{2+}$ signaling regulates the activation of CaMKK/ULK1, but not the nuclear translocation of TFEB. Inhibition of ULK1 by RNAi is found to increase intracellular survival of *Mtb*. Consistently, Horne et al. reported that ULK1 is genetically associated with *Mtb*, and ULK1-deficient cells exhibited lower autophagy and higher *Mtb* replication[59]. However, loss of ULK1 or other autophagy genes does not affect bacterial burdens during *Mtb* infection in vivo[19]. The functional role of autophagy genes in the restriction of *Mtb* replication is complicated by the fact that *Mtb* encode effective inhibitors of autophagy, as well as the additional roles of autophagy genes in the regulation of inflammatory responses[19,20,22]. Indeed, ULK1 has been shown regulate the expression of TNF[59]. Further study is required to clarify the functional role of CACNA2D3/ULK1 signaling axis in the pathogenesis of TB.

Individual miRNAs may target multiple transcripts rather than one specific gene. Although several transcripts including *Pink1* and *Plk2* were reported as miR-27a targets[40,41,60], the relevance between *miR-27a* and these genes need more validation. However, our finding showed that *Pink1* may not be required in the *miR-27a*-dependent regulation during *Mtb* infection. Also inhibition of *St6galnac3* or *Plk2* by RNAi attenuated the enhanced bacterial killing of *miR-27a-/-* macrophages totally or partially, respectively. Possibly, miR-27a may simultaneously target multiple transcripts including *Cacna2d3*, *Plk2* and *St6galnac3*, resulting in potent cumulative effects on regulating autophagy and *Mtb* viability.

Overall, our study have demonstrated that *Mtb* induce the expression of *miR-27a* to downregulate *Cacna2d3*, which thereby inhibiting $Ca^{2+}$-mediated autophagy, suggesting a strategy for *Mtb* to inhibit the host autophagy. Therefore, *Mtb* have evolved a series of successful strategies for inhibition and evasion of host autophagy. Indeed, the autophagy induction in response to *Mtb* infection is not so robust, which may shield the effect of deleting host autophagy genes or blocking autophagy on *Mtb* growth. So, the better way to examine the effect of autophagy on the *Mtb* infection is to disarm the anti-autophagy strategies of *Mtb*.

## Methods

**Clinical samples**. All protocols were approved by the local ethics committee of Tongji University School of Medicine (permit number: 2011-FK-03), and signed informed consent was obtained from all subjects. Diagnosis of TB was based on clinical presentation and radiological findings (such as an X-ray or computed tomography (CT) scan) and was confirmed by a positive sputum culture. The test for the anti-human immunodeficiency virus (HIV) antibody was negative for all TB patients. The average age was 40 years, and 65% of the patients were male.

The healthy control individuals came from physical examination donors. The inclusion criteria were no history of previous TB or anti-mycobacterial treatments and no evidence of TB-related infiltration in chest X-rays. The average age was 35 years, and 60% of the controls were male.

**Cell lines**. HEK293T cells and Raw264.7 cells were purchased from ATCC, and authenticated by the vendor. Atg5-/- Raw264.7 cell lines were gifted from Prof. Liu Cuihua's Lab of Institute of Microbiology, Chinese Academy of Sciences. All cells were mycoplasma-free with regular checks performed by a LookOut Mycoplasma PCR Detection Kit (MP0035, Sigma-Aldrich).

**Mice**. All animal study protocols were reviewed and approved by Tongji University School of Medicine review boards for animal studies.

*Cacna2d3PB/PB* mice were generated by PB transposon and gifted by Prof. Wu Xiaohui of Fudan University. Briefly, the PB transposon were inserted into the intron region between exon 27 and exon 28 of the *Cacna2d3* gene of FVB mice. *Cacna2d3* expression were disrupted due to the transcriptional termination signals within PB transposon[42–44]. *Cacna2d3PB/PB* mice were backcross to FVB background more than 10 generations and maintained in heterozygotes.

The *miR-27a-/-* mice were generated through the CRISPR/Cas9 method[33]. Briefly, in vitro-translated Cas9 mRNA and gRNA were co-microinjected into the C57BL/6 zygotes. The three gRNA sequences used to generate the knockout mice is GAGAAGCCTATCATGACAAC, CTGTGAACACGACTTTGCTG, and AGTGGCTAAGTTCCGCCCCC. Founders with frameshift mutations were screened with T7E1 assay and validated by DNA sequencing. Three F1 founders

were got for the *mir-27a[-/-]* mice, and one of them was chosen to backcross to the WT mice more than five generations to maintain the strain.

C57BL/6 mice were purchased from Shanghai Laboratory Animal Center (Shanghai, China). All mice were bred in specific pathogen-free (SPF) conditions at the Laboratory Animal Center of Tongji University. Female mice, 6–8 weeks-of-age, were used.

**Reagents and antibodies.** The VGCC inhibitor Bepridil (sc-202974) was purchased from Santa Cruz Biotechnology (Santa Cruz, USA).The vacuolar ATPase inhibitor Bafilomycin A1 (S1413) and CQ (S4157) was purchased from Selleck Chem. (Shanghai, China). SB203580(559389), SP600125(420119), PD98059 (513000), and PDTC(548000) were purchased from Calbiochem (Merck, Darmstadt, Germany) and were all used at a concentration of 10 μM. Adenovirus carrying mRFP-GFP-LC3B (C3011) or GFP-LC3B (C3006) were purchased from Beyotime Biotechnology (Shanghai, China).

Rabbit anti-CACNA2D3 (ab102939, 1:1000 for WB and 1:100 for IF) and mouse anti-Calnexin (ab31290, 1:100 for IF) were purchased from Abcam (Abcam, Cambridge, USA); rabbit anti-LC3A/B (4108, 1:1000 for WB and 1:100 for IF amd IHC), rabbit anti-TFEB (37785, 1:1000 for WB and 1:100 for IF), rabbit anti-ATG5 (12994, 1:1000), rabbit anti-p62 (8025, 1:1000), rabbit anti-Phospho-ULK1 (Ser555) (5869, 1:1000), rabbit anti-Phospho-ULK1 (Ser757) (14202, 1:1000), and rabbit anti-Phospho-CAMKII (Thr286) (12716, 1:1000) were purchased from Cell Signaling Technology (CST, CA, USA). Monoclonal mouse anti-GAPDH (SAB2701826, 1:1000) was purchased from Sigma-Aldrich (St. Louis, MO, USA). Monoclonal mouse anti-ACTINB (AC026, 1:5000) was purchased from Sigma-Aldrich (St. Louis, MO, USA).The HRP-conjugated anti-rabbit secondary antibody (W10804, 1:5000) was purchased from ThermoFisher Scientific (ThermoFisher Scientific, MA, USA). Hoechest33342 (H3570), Alexa Fluor488-conjugated anti-rabbit IgG (H + L) (A21206, 1:500 for IF) and Alexa Fluor 555-conjugated anti-mouse IgG (H + L) (A32727, 1:500 for IF) were purchased from ThermoFisher Scientific.

**PMBCs isolation.** PBMCs were isolated from EDTA-treated whole blood through density gradient centrifugation by Ficoll separation (Ficoll-Paque plus; Amersham Biosciences). After two washes in PBS, the PBMCs were subjected to further analyses.

**Mouse macrophage isolation and infection.** Peritoneal macrophages were isolated as described[61]. Briefly, mice were injected intraperitoneally (IP) with 2.0 mL of 4% Brewer's thioglycollate medium (B2551, Sigma-Aldrich). After 3 days, primary macrophages in the peritoneal lavage were collected from euthanized animals using 10 mL cold PBS. Subsequently, the cells were plated in 12-well plates at 10[6] cells/well in RPMI 1640 supplemented with 10% FBS and penicillin/streptomycin and were incubated at 37 °C in 5% CO$_2$ for 2 h. The cultures were then washed three times with PBS to remove non-adherent cells, and the remaining adherent monolayer cells were used as primary peritoneal macrophages.

*Mtb* H37Rv (H37Rv) were grown to mid-log phase in Middle brook 7H9 medium (271310, Becton Dickinson, Cockeysville, MD) with 0.05% Tween-80 and 10% oleic acid-albumin-dextrose-catalase (OADC) enrichment (211886, Becton Dickinson, Sparks, MD). Before infection, *Mtb* were suspended in complete medium without antibiotics. Peritoneal macrophages were infected with H37Rv at a MOI of 5.

**Calcium ion flux measurement.** To quantify the ion fluxes, fluorometry was measured in Cellomics plate reader (Thermo Fisher Scientific, MA, USA). For calcium flux measurement, the Fluo-4 NW calcium assay kit (Invitrogen) was used. Calcium fluxes were measured in 20,000 macrophages plated in a 96-well plate.

**Immunofluorescence staining and confocal microscopy analysis.** All images were captured on a Leica SP5 confocal microscope at ×63 magnification. For CACNA2D3 and TFEB staining, the primary peritoneal macrophages were fixed with 4% formaldehyde overnight, permeabilized with 0.1% Triton X-100 in PBS for 10 min, and blocked with 3% BSA in PBS for 30 min at RT. The cells were stained with the indicated antibodies at a dilution of 1:200 in 3% BSA in PBS for 6 h at 4 °C and were then incubated with Invitrogen Alexa Fluor 488 or 555 conjugated secondary antibodies at a dilution of 1:1000 for 2 h at RT. The nuclei were stained with Hoechst 33342.

For mRFP-GFP-LC3B and GFP-LC3B analysis, primary peritoneal macrophages were pre-treated with adenovirus carrying mRFP-GFP-LC3B or GFP-LC3B for 48 h before further treatment. Then, the cells were imaged using confocal microscopy. LC3B spots were counted using the Cellomics ArrayScan platform.

**Luciferase assay.** A 0.5-kb region of the mouse *Cacna2d3* 3′-UTR containing the predicted miRNA miR-27a-binding sites was cloned into psi-CHECK2 to obtain the *Cacna2d3* 3′ UTR-luc construct. The *Cacna2d3* 3′-UTR (mut)-luc construct was obtained from the *Cacna2d3* 3′-UTR construct by mutating the complementary sequence of the miR-27a seed region (5′-ACUGUGA-3′ to 5′-AAGG ACA-3′). The constructs were co-transfected with synthesized dsRNAs mimicking

the miR-27a miRNA or mimic NCs (5′-UUCUCCGAACGUGUCACGUUT-3′) into HEK293T cells, and the luciferase assay was performed using the dual-luciferase reporter assay system (Promega, Madison, USA).

**Aerosol infection of mice.** All animal study protocols were reviewed and approved by Tongji University School of Medicine review boards for animal studies. Infection studies were carried out using a murine respiratory infection model[62–64]. Mice were infected with ~200 CFU of *Mtb* H37Rv using a Glas-Col inhalation exposure system (Glas-Col Inc., Terre Haute, Ind.). At 28 days post infection, the mice were sacrificed and bacterial counts in the lungs were determined by plating 10-fold serial dilutions of individual lung homogenateson Middlebrook 7H10 agar supplemented with 10% Middlebrook OADC enrichment. *Mtb* colonies were incubated at 37 °C and were counted after 21 days.

For miR-27a intervention, a miR-27a antagomir (5′-GCGGAACUUAGCCACUGUCAA-3′) and a NC antagomir were purchased from RIBOBIO (Guangzhou, China). Entranster TM-in vivo transfection reagent was ordered from Engreen® (Beijing, China). The transfection protocol was modified based on manufacturer's instructions. Briefly, reagent A was first prepared by dissolving 50 nmol of miR-27a antagomir or the NC antagomir in 375 μL of autoclaved ddH$_2$O, and then 375 μL of a sterile 10% glucose solution was added and mixed well. Next, reagent B was prepared by adding 375 μL of sterile 10% glucose solution to 375 μL of Entranster TM-in vivo transfection reagent, which was then mixed well. Finally, reagents A and B were mixed (1:1) to yield the working solution. For each injection, we used 300 μL of the working solution (~10 nmol of the miR-27a antagomir/NC antagomir).

**Transmission electron microscopy.** Primary peritoneal macrophages from WT FVB and *Cacna2d3* mutant mice were infected with *Mtb* H37Rv strains at an MOI of 5 for 6 h. Fixation and sectioning were performed using standard procedures by the Microscopy Core at Shanghai Jiaotong University Medical School. Photographs were captured using HITACHI H-7650, and at least 20 images were acquired for each structure of interest; representative images are shown.

**Single cell suspensions of *Mtb*.** Briefly, the H37Rv culture was dispersed by aspiration five times each with a 23-gauge and then a 26-gauge needle, followed by an additional dispersion by aspiration three times through a 30-gauge needle. The dispersed bacteria were allowed to stand for 5 min to allow the clumps to settle down. The upper half of the suspension was then used for the experiments. Quantitation of bacteria was done by taking absorbance at 600 nm wavelength[65] (0.6 OD corresponds to ~100 × 10[6] bacteria).

**Histological analysis and acid-fast staining.** Lung tissues from *Mtb*-infected mice were fixed in 4% phosphate-buffered formalin for 24 h and were then embedded in paraffin wax. The paraffin-embedded lungs were cut in serial sections with a thickness of 2–3 μm. Hematoxylin and eosin (H&E) staining were applied to detect the infiltration in the lungs. Acid-fast staining with the standard Ziehl–Neelsen method was employed for the determination of *Mtb* burden in the lungs. Five noncontiguous sections were examined for each lung, and at least five animals were sacrificed for each group. The stained slides were visualized by light microscopy.

For immunohistochemistry, lung tissue from three PTB patients with pulmonary lobectomy and six *Mtb*-infected mice was investigated. Segments of lung tissues were fixed in 10% buffered formalin and were embedded in paraffin. The blocks were cut into 5 μm sections, and five noncontiguous sections were processed for IHC staining with LC3B antibody at a 1:250 dilution. The sections were then incubated with the Supersensitive 1-step polymer-HRP detection system (Biogenex). Immunostaining was visualized with 3,3-diaminobenzidine (DAB, Biogenex) substrate. After counterstaining in hematoxylin, the sections were mounted and examined. The studies on clinical samples were conducted in accordance with ethical guidelines of the Institutional Review Board of Tongji University.

**RNA interference and transfection.** Mouse peritoneal macrophages were transfected with the small interference RNA, microRNA mimic or inhibitor with INTERFER in the transfection reagent (AM4511, ThermoFisher). The siRNAs specifically targeting mouse *Cacna2d3* and *Tfeb* were purchased from Santa Cruz. The siRNAs pool specifically targeting mouse *Plk2*, *St6galnac3*, and *Pink1* were synthesized from Genepharma Co., Ltd (Shanghai, China) and the sequence are listed in Supplementary Table 2. The miR-27a mimics and inhibitors were purchased from RIBOBIO (Guangzhou, China).

**Q-PCR.** For detection of *Cacna2d3*, total RNA from PBMCs and mice macrophages and lungs was extracted with TRIzol reagent according to the manufacturer's instructions (Invitrogen). One microgram of total RNA was reverse transcribed using the ReverTra Ace® qPCR RT Kit (Toyobo, FSQ-101) according to the manufacturer's instructions. A SYBR RT-PCR kit (Toyobo, QPK-212) was used for quantitative real-time PCR analysis. The relative mRNA expression of different genes was calculated by comparison with the control gene *Gapdh* (encoding

GAPDH) using the $2^{-\triangle\triangle Ct}$ method. The sequences of primers for the qPCR analysis are shown in Supplementary Table 1.

For the detection of miR-27a, miR-23a, and miR-24, PBMCs and mice macrophages and lungs were collected in TRIzol (Invitrogen) and were performed using the standard procedure. The miRNeasy Mini kit (QIAGEN) and TaqMan MicroRNA Reverse Transcription kit (Applied Biosystems) were used for total RNA and cDNA preparation, respectively. Power SYBR Green master mix (Applied Biosystems) was used for the qRT-PCR analysis on a 7900HT Fast Real-Time PCR System (Applied Biosystems). Sno-RNA U18 was used as an internal control. One-tailed $t$-tests were performed with GraphPad Prism 7.0, and a $P$-value $\leq 0.05$ was considered significant. The sequences of primers were purchased from Tiangen Co., Ltd (Shanghai, China) and the Cat. no. are shown in Supplementary Table 1.

**Western blot**. Proteins were lysed from cells using RIPA buffer containing 10 mM Tris–Cl, pH 8.0, 150 mM NaCl, 1% Triton X-100, 1% Na-deoxycholate, 1 mM EDTA, 0.05% SDS and fresh 1 × proteinase inhibitor. The protein concentration was determined via the Bradford method using the Bio-Rad protein assay before proteins were equally loaded and separated in polyacrylamide gels. The proteins were then transferred to a nitrocellulose filter membrane (Millipore) and were incubated overnight with indicated primary antibodies. HRP-conjugated secondary antibodies were then applied to the membrane, and the Western blot signal was detected using auto-radiographic film after incubation with ECL (GE Healthcare) or SuperSignal West Dura reagents (Thermo Scientific).

**Statistical analysis**. The results are represented as the mean ± s.e.m., and statistical significance between groups was determined using an unpaired $t$-test or the Mann–Whitney $U$-test. GraphPad Prism software 7.0 was used for all analyses, and a *$p < 0.05$ was considered statistically significant.

## Data availability

Data have been deposited in the Gene Expression Omnibus under accession code GSE119494, GSE119495, GSE119496. Other data that support the findings of this study are available from the corresponding author upon request.

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

## Acknowledgements

We thank Prof. Xu Tian and Prof. Wu Xiaohui for the *Cacna2d3* mutant mice. We thank Prof. Liu Cuihua for the Atg5[-/-] Raw264.7 cells. We thank Feifan Xu (Nantong Sixth people's Hospital, China) for samples collection. We acknowledge Dr. Xinchun Chen (Shenzhen Third people's Hospital) and members of the Xinchun Chen's laboratory for expert technical assistance of mice infection. We thank Dr. Gucheng Zeng (Sun Yat-Sen University) and members of his laboratory for expert technical assistance of mice infection. We thank members of the Shanghai Key Lab of Tuberculosis and Baoxue Ge's laboratory for helpful discussions. This work was supported by National Natural Science Foundation of China (Nos. 81300003, 81501373, 91542111, and 81370108).

## Author contributions

F.L., J.X.C., and B.X.G. designed experiments. J.X.C., F.L., P.W., H.H.L. Y.L.Z., H.P.L., Z.H.L., and R.J.Z. performed experiments and analyzed data. P.W. facilitated clinical sample collection and transport. X.C.H. and W.J. prepared the bacteria H37Rv. R.J.Z., H.Z.L., F.W., and J.X.C. contributed to mouse experiments. J.X.C., F.L., Y.L.Z., H.P.L., H.H.L., and B.X.G. prepared the figures. F.L., J.X.C., and B.X.G. wrote the manuscript.
