## [Peer Review File · Nature Communications]

Reviewers' Comments:

Reviewer #1:

Remarks to the Author:

Liu et al reported that miR-27a is highly expressed in active TB patients, in the lung of Mycobacterium tuberculosis (Mtb)-infected mice, and in Mtb-infected macrophages. They showed that miR-27a inhibits autophagy by suppressing the expression of its target gene CACNA2D3 (Cac3), a calcium transporter located on the ER. By generating and analyzing miR-27a^{-/-} and Cac3 mutant mice (Cac3^{m/m}), as well as their macrophages, they showed that miR-27 facilitates Mtb survival, while Cac3 promotes host control of Mtb. Furthermore, the authors showed that in Cac3^{m/m} mice and macrophages, the effect of miR-27a inhibitors and mimics on autophagy and Mtb is abrogated, suggesting that miR-27a controls autophagy and Mtb survival through Cac3 and calcium signaling.

The finding that Mtb induces miR-27a expression to facilitate the former's escape from autophagy-mediated killing by suppressing Cac3-mediated calcium signaling, which was previously shown to promote autophagy, is novel and interesting. However, there are a few concerns that need to be addressed:

1. To what degree do Cac3 and calcium signaling mediate the effect of miR-27a on autophagy and Mtb survival? An easy genetic test is to generate miR-27a^{-/-};Cac3^{m/+} and miR-27a^{-/-};Cac3^{m/m} mice and to assess whether restoring Cac3 expression to WT levels is sufficient to negate the effects of miR-27a-deficiency. A few results presented in this manuscript suggest that the functional importance of Cac3 and calcium signaling in mediating miR-27a effect may be more modest than the authors would like to suggest:

a. Fig. 3k,l: while Cac3 mRNA and protein are expressed at very low levels in WT lung and macrophages, they are drastically increased in miR-27a^{-/-} lung and macrophages. Somehow this drastic increase in Cac3 expression did not translate into a strong effect on calcium signaling (Fig. 6f). This is surprising considering that calcium signaling is completely abolished in Cac3^{m/m} macrophages (Fig. 4i). One has to argue that the trace amount of Cac3 in WT macrophages is sufficient for its function and any further increase in the cellular concentration of Cac3 protein doesn't significantly enhance calcium signaling. If this is true, pathways other than Cac3 and calcium must be invoked to explain the significant effect of miR-27a deficiency on autophagy (Fig. 1j,k,l) and Mtb control (Fig. 2).

b. Fig. 6d, the right most two bars: in Cac3^{m/m} macrophages, miR-27a inhibitor still significantly promoted autolysosome formation, suggesting that the effect of miR-27a inhibitor can be mediated by pathways other than Cac3 and calcium signaling.

Minor concerns:

2. As miR-27a exists in the miR-23a/27a/24-2 cluster, deletion of miR-27a may affect the expression of the other two miRNAs in the cluster, i.e. miR-23a and miR-24. The expression levels of miR-23a and miR-24 should be examined in miR-27a^{-/-} lung and macrophages, and compared with their WT counterparts.

3. Fig. 4c: TEM analysis needs quantification.

4. Fig. 5d: The immunohistochemistry of LC3B is hard to interpret. Does blue staining indicate LC3B? If so, why did the Cac3^{m/m} lung show more LC3B staining?

5. Fig. 6e: is the color code labeled correctly? The text and figure do not match.

Reviewer #2:

Remarks to the Author:

Not much is known about the regulation of calcium-dependent autophagy mechanisms in either normal physiology or pathophysiology. Liu et al show that Mycobacterium tuberculosis (Mtb) infection induces a miRNA (mir-27a) in human and mouse cells. Via a novel target, an ER-resident calcium channel containing the CACNA2D3 polypeptide, this results in suppression of anti-microbial autophagy in cell models and in mouse models of Mtb infection. This is a very interesting finding with likely clinical relevance too, with mir-27a upregulated in human cases of tuberculosis and antagomirs of mir-27a diminishing infection burden in mouse models of Mtb infection.

Generally, speaking the manuscript is well written with little redundancy with existing published findings. Most experiments are well-designed leading to mostly robust conclusions. The methods are mostly well-documented, enabling reproducibility. There are nonetheless a number of specific instances where these standards are not reached (see specific points below). These should be addressed prior to publication.

Also, the authors might examine the role of mir27a in repression of PINK1, a regulator of mitochondrial autophagy and a known target of mir27a (Molecular Neurodegeneration 2016 11:55). This is because Parkin, the partner of PINK1 in mitophagy, has also been suggested to be involved in stimulating autophagy of Mtb.

Major points

- 1) How many founders were used for the mir27a^{-/-} mice? Have they been backcrossed to exclude off-target effects of the gRNA in the first generations? This information is needed to interpret properly the validity of the data using this model. Similarly, information on the strain background of the Cacna2d3 mice, the breeding program to maintain homozygotes and what the strain background of the wild-type controls used to compare these homozygotes against is required. Also, what is the targeted gene mutation in Cacna2d3 - what does it do the the function of this gene?
- 2) Lines 135-138. Experiments with pleiotropic autophagy blocking drugs should be performed with, for example, Atg5^{-/-} macrophages to ensure the apparent redundancy with mir27a of autophagy is really autophagy and not some other target of these drugs
- 3) Proper statistical treatment of in vivo infection (CFU assay) experiments is missing from figures throughout. This is a presumed oversight? Statistical thresholds used for these experiments are indeed given in the figure legends...
- 4) Figure 1 – it does appear that autophagic flux is increased with mir27a ablation. It is not possible to say that this is due to increased autophagosome maturation per se. A true flux assay, biochemically (LC3B-II blot) or by immunofluorescence (tandem LC3 reporter), such as a time course in the presence versus absence of chloroquine or bafilomycin is required to resolve this. This is worth doing as the regulation of autophagy by mir27a^{-/-} is a fundamental observation in this study that requires complete characterisation. Similar comments apply to the later flux assays performed where Cacna2d3 levels are manipulated.
- 5) In addition to point 4, the authors should exclude that the association of Mtb with autophagosomes (albeit forming at a reduced rate due to mir27a) is not disrupted by mir27a (i.e. abundance of Mtb-positive autophagosomes should reduce proportionally with overall reduction in autophagosomes).
- 6) It would be useful to present controls for the efficacy of mir-27a antagomir on mir27 levels/function in the in vitro and in vivo experiments (i.e. known rather than novel mir27 targets described here or other readout?). This could then enable the reader to conclude that the effects of the antagomir are likely to be “on-target” effects.
- 7) In the TEM analysis of Cac3m/m macrophages (lines 237), are these bacteria unincorporated into autophagosomes? Is there a difference observable between these and the proportion of Mtb +ve autophagosomes in the TEM from wild-type mice?
- 8) Figure 4i – Mtb infection perturbs calcium levels within seconds, as shown, but effects on

autophagy and viability are on the time scale of hours. What is the effect on calcium levels at such later time points where evidence of autophagy regulation can actually be obtained?

9) Figure 5. The pathology of lungs in the infection model is presented largely without any quantification. Are these changes quantitatively significant between the genotypes?

Minor points

1) Define PBMCs first time acronym is used (line 94)

2) Why did the authors choose to focus on mir27a after the initial expression analyses (Figure 1a-c)? An explanatory note to this effect would be good.

3) Lines 111-112. Use of these chemical inhibitors does not conclusively demonstrate a TLR2/ERK/NfkappaB signaling pathway. This should be acknowledged in text.

4) Lines 118-124. For clarity, the authors should be explicit about how they conclude the initial infection/phagocytosis of Mtb is unaffected but the subsequent survival is affected (i.e. analysis of different time points within the data).

5) Line 135 – “influx” is not correct, “flux” is the correct term.

6) Please check that panels 4a and 4b are in the correct order and referenced so in the text and Figure legends.

7) Please check panel references of statistical thresholds at end of legend for Figure 6. These may be mislabelled.

Reviewer #3:

Remarks to the Author:

This manuscript describes the regulation of autophagy by Mycobacterium tuberculosis via microRNA (miRNA) miR-27a and an endoplasmic reticulum calcium transporter. This is a novel finding providing new insight to the intracellular survival of the tubercule bacillus. The work is exciting because M. tuberculosis encodes a number of proteins known to bind calcium. The authors provide a compelling story that convincingly shows this novel regulatory pathway. I found the work to be comprehensive and very well done. I recommend the acceptance of this paper.

Reviewer #1 (Remarks to the Author):

Liu et al reported that miR-27a is highly expressed in active TB patients, in the lung of Mycobacterium tuberculosis (Mtb)-infected mice, and in Mtb-infected macrophages. They showed that miR-27a inhibits autophagy by suppressing the expression of its target gene CACNA2D3 (Cac3), a calcium transporter located on the ER. By generating and analyzing miR-27a^{-/-} and Cac3 mutant mice (Cac3^{m/m}), as well as their macrophages, they showed that miR-27 facilitates Mtb survival, while Cac3 promotes host control of Mtb. Furthermore, the authors showed that in Cac3^{m/m} mice and macrophages, the effect of miR-27a inhibitors and mimics on autophagy and Mtb is abrogated, suggesting that miR-27a controls autophagy and Mtb survival through Cac3 and calcium signaling. The finding that Mtb induces miR-27a expression to facilitate the former's escape from autophagy-mediated killing by suppressing Cac3-mediated calcium signaling, which was previously shown to promote autophagy, is novel and interesting. However, there are a few concerns that need to be addressed:

We thank the reviewer for appreciating the novelty of our current work, and will address the concerns as following.

1. To what degree do Cac3 and calcium signaling mediate the effect of miR-27a on autophagy and Mtb survival? An easy genetic test is to generate miR-27a^{-/-};Cac3^{m/+} and miR-27a^{-/-};Cac3^{m/m} mice and to assess whether restoring Cac3 expression to WT levels is sufficient to negate the effects of miR-27a-deficiency. A few results presented in this manuscript suggest that the functional importance of Cac3 and calcium signaling in mediating miR-27a effect may be more modest than the authors would like to suggest:

We appreciate the reviewer's comments. Our current data showed that the effect of the miR-27a mimic or inhibitor on the survival of *Mtb* or the formation of LC3 puncta is observed in wild type (WT) macrophages, but not in *Cac3*^{PB/PB} macrophages (**Fig. 6a-d, Supplementary Fig. 13a, b**), suggesting that *Cacna2d3* may mediate the miR-27a's effect in regulating the process of *Mtb* infection.

However, we fully agree with the reviewer that generation of *miR-27a*^{-/-} *Cac3*^{PB/PB} mice to assess whether restoring *Cacna2d3* expression to WT levels is sufficient to negate the effects of miR-27a-deficiency would be much helpful to strengthen our finding that *Cacna2d3* mediates the effect of miR-27a on autophagy and *Mtb* survival. We have tried very hard to generate *miR-27a*^{-/-} *Cac3*^{PB/PB} mice for a long time, but haven't got it so far, probably due to the low fertility.

a. Fig. 3k,l: while Cac3 mRNA and protein are expressed at very low levels in WT lung and macrophages, they are drastically increased in miR-27a^{-/-} lung and macrophages. Somehow this drastic increase in Cac3 expression did not translate into a strong effect on calcium signaling (Fig. 6f). This is surprising considering that calcium signaling is completely abolished in Cac3^{m/m} macrophages (Fig. 4i).

One has to argue that the trace amount of Cac3 in WT macrophages is sufficient for its function and any further increase in the cellular concentration of Cac3 protein doesn't significantly enhance calcium signaling. If this is true, pathways other than Cac3 and calcium must be invoked to explain the significant effect of miR-27a deficiency on autophagy (Fig. 1j, k, l) and Mtb control (Fig. 2).

Thank the reviewer for the illuminating question. *Cacna2d3* encodes a regulatory $\alpha_2\delta$ subunit of voltage gated calcium channels, which is required for the assembly of the channel on the membrane. *Cacna2d3*'s effect on calcium signaling may be modest, most likely due to the loose association of $\alpha_2\delta$ subunit with the channel (Cassidy *et al.*, 2014, Zamponi *et al.*, 2015), and a possible saturation of channel assembly on the membrane. That may explain why increase in the cellular concentration of *Cacna2d3* protein only lead to a ~ 2-fold increase of calcium signaling (Fig. 6e, f). Furthermore, we agree with the reviewer's point that *miR-27a* may have multiple targets other than *Cacna2d3* and Ca^{2+} signaling (Tian *et al.*, 2014; Kim *et al.*, 2016; Zhang *et al.*, 2017).

b. Fig. 6d, the right most two bars: in *Cac3*^{m/m} macrophages, miR-27a inhibitor still significantly promoted autolysosome formation, suggesting that the effect of miR-27a inhibitor can be mediated by pathways other than Cac3 and calcium signaling.

We agree with the reviewer's point. Besides *Cacna2d3*, miR-27a has multiple targets including *Plk2*, *Pink1*, *Fas* and *Scd1* (Tian *et al.*, 2014; Kim *et al.*, 2016; Zhang *et al.*, 2017). According to our results, we concluded that *miR-27a* may regulate autophagy and intracellular survival of *Mtb* at least partially through *Cacna2d3*-mediated Ca^{2+} signaling.

Minor concerns:

2. As miR-27a exists in the miR-23a/27a/24-2 cluster, deletion of miR-27a may affect the expression of the other two miRNAs in the cluster, i.e. miR-23a and miR-24. The expression levels of miR-23a and miR-24 should be examined in miR-27a^{-/-} lung and macrophages, and compared with their WT counterparts.

As per the reviewer's suggestion, we have analyzed the expression of miR-23a and miR-24 in the macrophages isolated from *miR-27a*^{-/-} mice, and compared them with their WT counterparts. The data demonstrated that the expression levels of *miR-23a* and *miR-24* show no significant difference between *miR-27a*^{-/-} and WT macrophages. We have added this part in the main text (Supplementary Fig. 2d).

3. Fig. 4c: TEM analysis needs quantification.

As per the reviewer's suggestion, we have provided the quantification of TEM analysis in the revised manuscript (Supplementary Fig. 8e).

4. Fig. 5d: The immunohistochemistry of LC3B is hard to interpret. Does blue staining indicate LC3B? If so, why did the Cac3m/m lung show more LC3B staining?

The brown staining indicated LC3B. According to our results, there is less LC3B staining in the lung tissues of *Cac3^{PB/PB}* mice as compared to the WT mice. As per the reviewer's suggestion, we have added this part to the figure legend of revised manuscript.

5. Fig. 6e: is the color code labeled correctly? The text and figure do not match.

We thank the reviewer for the correction. The labels of the color code have been corrected in the revised manuscript (Fig. 6e).

Reviewer #2 (Remarks to the Author):

Not much is known about the regulation of calcium-dependent autophagy mechanisms in either normal physiology or pathophysiology. Liu et al show that Mycobacterium tuberculosis (Mtb) infection induces a miRNA (mir-27a) in human and mouse cells. Via a novel target, an ER-resident calcium channel containing the CACNA2D3 polypeptide, this results in suppression of anti-microbial autophagy in cell models and in mouse models of Mtb infection. This is a very interesting finding with likely clinical relevance too, with mir-27a upregulated in human cases of tuberculosis and antagomirs of mir-27a diminishing infection burden in mouse models of Mtb infection. Generally, speaking the manuscript is well written with little redundancy with existing published findings. Most experiments are well-designed leading to mostly robust conclusions. The methods are mostly well-documented, enabling reproducibility. We thank the reviewer for pertinent comments and appreciating for the novelty and quality of our study.

There are nonetheless a number of specific instances where these standards are not reached (see specific points below). These should be addressed prior to publication.

As per the reviewer's suggestion, we have addressed the reviewer's concern as described below.

Also, the authors might examine the role of mir27a in repression of PINK1, a regulator of mitochondrial autophagy and a known target of mir27a (Molecular Neurodegeneration 2016 11:55). This is because Parkin, the partner of PINK1 in mitophagy, has also been suggested to be involved in stimulating autophagy of Mtb.

As per the reviewer's suggestion, we have analyzed the effect of *Pink1*, a regulator of mitochondrial autophagy and a known target of miR-27a (Kim *et al.*, 2016), on intracellular survival of *Mtb* in WT and *miR-27a*^{-/-} macrophages. The data showed that inhibition of *Pink1* by RNAi significantly decrease the intracellular survival of

Mtb in both of WT and *miR-27a*^{-/-} macrophages as determined by CFU assay, indicating *Pink1* as a positive regulator of *Mtb* survival. Therefore, the enhanced effect of *miR-27a* on the intracellular survival of *Mtb* is unlikely through downregulating *Pink1*. We have added this part into the main text (**Supplementary Fig. 13d, e**).

Major points

1) How many founders were used for the *mir27a*^{-/-} mice? Have they been backcrossed to exclude off-target effects of the gRNA in the first generations? This information is needed to interpret properly the validity of the data using this model. Similarly, information on the strain background of the *Cacna2d3* mice, the breeding program to maintain homozygotes and what the strain background of the wild-type controls used to compare these homozygotes against is required. Also, what is the targeted gene mutation in *Cacna2d3* - what does it do the function of this gene?

Thank the reviewer for the question. We have obtained 3 F1 founders for the *miR-27a*^{-/-} mice, and chosen one of them to backcross to the WT mice more than 5 generations to maintain the strain. The information was added to the methods section.

Thank the reviewer for the question. The *Cacna2d3* mutant mice were generated using piggyback (PB) transposon in FVB strains, and were gifted by Prof. Xiaohui Wu from Fudan University (**Ding et al., 2005, 2014**). Briefly, PB transposon was inserted into the intron between exon 27 and exon 28 of the *Cacna2d3* gene loci, and the expression of *Cacna2d3* is subsequently disrupted (**Supplementary Fig. 8b**), due to the insertion of transcriptional termination signals within PB transposon (**Shi et al., 2014; Wang et al., 2016; Cui et al., 2016; Zhu et al., 2017**). *Cacna2d3* mutant mice were backcross to FVB background more than 10 generations and maintained in heterozygotes.

To be more precisely, we have revised the description of the *Cacna2d3* mutant mice as *Cac3*^{PB/PB} in the whole manuscript.

2) Lines 135-138. Experiments with pleiotropic autophagy blocking drugs should be performed with, for example, *Atg5*^{-/-} macrophages to ensure the apparent redundancy with *mir27a* of autophagy is really autophagy and not some other target of these drugs.

As per the reviewer's suggestion, we have applied *Atg5*^{-/-} macrophages and confirmed that the altered viability effect of *miR-27a* mimic or inhibitor on the *Mtb* infection is truly through autophagy (**Supplementary Fig. 3c, d**).

3) Proper statistical treatment of in vivo infection (CFU assay) experiments is missing from figures throughout. This is a presumed oversight? Statistical thresholds used for these experiments are indeed given in the figure legends...

As per the reviewer's suggestion, we have added the statistics treatment and included the thresholds of *in vivo* infection experiments into the figures (Fig. 2c, 2f, 5c and 6m).

4) Figure 1 – it does appear that autophagic flux is increased with mir27a ablation. It is not possible to say that this is due to increased autophagosome maturation per se. A true flux assay, biochemically (LC3B-II blot) or by immunofluorescence (tandem LC3 reporter), such as a time course in the presence versus absence of chloroquine or bafilomycin is required to resolve this. This is worth doing as the regulation of autophagy by mir27a^{-/-} is a fundamental observation in this study that requires complete characterisation. Similar comments apply to the later flux assays performed where *Cacna2d3* levels are manipulated.

As per the reviewer's suggestion, we have examined the mRFP-GFP-LC3 reporter pattern in the presence versus absence of chloroquine to carefully determine the effect of *mir-27a* on autophagy flux. The results showed that in the presence of chloroquine (CQ) (Ouimet *et al.*, 2016), deficiency of miR-27a dramatically increase autophagosome formation and LC3 activation, suggesting that the increased autophagic flux with miR-27a ablation is due to increased autophagosome maturation per se. We have added this part into the main text (Supplementary Fig. 4a-c). Also, our results showed that in the presence of CQ, deficiency of *Cacna2d3* lead to dramatically decreased LC3-II amount during *Mtb* infection. (Supplementary Fig. 8f)

5) In addition to point 4, the authors should exclude that the association of *Mtb* with autophagosomes (albeit forming at a reduced rate due to mir27a) is not disrupted by mir27a (i.e. abundance of *Mtb*-positive autophagosomes should reduce proportionally with overall reduction in autophagosomes).

According to our results, *miR-27a* was shown to suppress both autophagosome formation and the *Mtb*-autophagosome co-localization, but overall reduction in autophagosomes appears to be very close to *Mtb*-positive autophagosomes (Fig. 1i, Supplementary Fig. 14d). These results suggested that the increased *Mtb*-positive autophagosomes in *miR-27a*^{-/-} macrophages may at least partially result from the enhanced autophagosome formation.

6) It would be useful to present controls for the efficacy of mir-27a antagomir on mir27 levels/function in the in vitro and in vivo experiments (i.e. known rather than novel mir27 targets described here or other readout?). This could then enable the reader to conclude that the effects of the antagomir are likely to be “on-target” effects.

As per the reviewer's suggestion, we have analyzed the expression pattern of some known *miR-27a* targets such as *Plk2* and *Pink1* as controls in the lung tissues of mice treated with *miR-27a* antagomir to ensure the efficacy of antagomir (Tian *et al.*, 2014; Kim *et al.*, 2016). The results showed that all of these genes' expression are

up-regulated in the lung tissues of *miR-27a* antagomir-treated mice, suggesting that the observed effects of *miR-27a* antagomir are most likely to be “on-target” effects. We have added this part into the main text (**Supplementary Fig. 5e**).

7) In the TEM analysis of $Cac3^{m/m}$ macrophages (lines 237), are these bacteria unincorporated into autophagosomes? Is there a difference observable between these and the proportion of *Mtb* +ve autophagosomes in the TEM from wild-type mice?

In our experiments, WT and $Cac3^{PB/PB}$ macrophages were infected with H37Rv for 24 hours and then sent to TEM analysis. At this time-point, most of the H37Rv bacteria were not likely in autophagosomes, but present in lysosome or cytoplasm (**Rahman *et al.*, 2014**).

8) Figure 4i – *Mtb* infection perturbs calcium levels within seconds, as shown, but effects on autophagy and viability are on the time scale of hours. What is the effect on calcium levels at such later time points where evidence of autophagy regulation can actually be obtained?

Generally, Ca^{2+} signaling is a short-term signaling which is activated within seconds. However, Ca^{2+} ions usually functions through their binding to calmodulins, which in turn activate the downstream effectors, thus maintaining the signaling for hours. (**Berg *et al.*, 2015**)

9) Figure 5. The pathology of lungs in the infection model is presented largely without any quantification. Are these changes quantitatively significant between the genotypes?

As per the reviewer’s suggestion, we have provided the quantification of histopathology results as shown in **Supplementary Fig. 4e**. The data showed significant differences of the histopathological impairments between genotypes.

Minor points

1, Define PBMCs first time acronym is used (line 94)

As per the reviewer’s suggestion, we have defined PBMCs as “peripheral blood mononuclear cell” in the main text.

2, Why did the authors choose to focus on *mir27a* after the initial expression analyses (Figure 1a-c)? An explanatory note to this effect would be good.

As *miR-27a*’s functional role in the regulation of *Mtb* infection remains uncharacterized, we choose it for our further study. As per the reviewer’s suggestion, we have added an explanatory note in the revised manuscript.

3, Lines 111-112. Use of these chemical inhibitors does not conclusively demonstrate a TLR2/ERK/NfkappaB signaling pathway. This should be acknowledged in text.

In our study, selective inhibition of ERK MAPK signaling pathway by PD98059 or NF- κ B signaling pathway by PDTC significantly suppressed *Mtb*-induced miR-27a's expression (**Supplementary Fig. 1f**), suggesting that *Mtb* infection may induce miR-27a expression via the activation of ERK MAPK signaling pathways and NF- κ B signaling pathways. As per the reviewer's suggestion, we have edited this part accordingly in the revised manuscript.

4, Lines 118-124. For clarity, the authors should be explicit about how they conclude the initial infection/phagocytosis of *Mtb* is unaffected but the subsequent survival is affected (i.e. analysis of different time points within the data).

Thank the reviewer for instruction. For our CFU assay, we have measured the CFU at 2 hours to assure the uptake of *Mtb* is equal (**Yang *et al.*, 2016**). The data showed that treatment of *miR-27a* mimic or inhibitor has no significant effect on the phagocytosis of *Mtb* in macrophages. (**Fig. 1d, e and Supplementary Fig. 2a**)

5, Line 135 – “influx” is not correct, “flux” is the correct term.

As per the reviewer's suggestion, we have changed the “influx” to “flux” in our revised manuscript.

6, Please check that panels 4a and 4b are in the correct order and referenced so in the text and Figure legends.

Thank the reviewer for correction, we have re-ordered these two panels in the revised manuscript (**Fig. 4a, b**).

7, Please check panel references of statistical thresholds at end of legend for Figure 6. These may be mislabelled.

Thank the reviewer for correction. We have re-labeled them in the revised manuscript.

Reviewer #3 (Remarks to the Author):

This manuscript describes the regulation of autophagy by *Mycobacterium tuberculosis* via microRNA (miRNA) miR-27a and an endoplasmic reticulum calcium transporter. This is a novel finding providing new insight to the intracellular survival of the tubercule bacillus. The work is exciting because *M. tuberculosis* encodes a number of proteins known to bind calcium. The authors provide a compelling story that convincingly shows this novel regulatory pathway. I found the work to be comprehensive and very well done. I recommend the acceptance of this paper.

We thank the reviewer for appreciating the novelty and quality of our research work.

References

1. Cassidy, J.S., Ferron, L., Kadurin, I., Pratt, W.S. & Dolphin, A.C. Functional exofacially tagged N-type calcium channels elucidate the interaction with auxiliary $\alpha 2\delta$ -1 subunits. *Proc Natl Acad Sci USA* **111**, 8979–8984 (2014).
2. Gerald, W., Zamponi, J.S., Alexandra, K. & Annette, C.D. The Physiology, Pathology, and Pharmacology of Voltage-Gated Calcium Channels and Their Future Therapeutic Potential. *Pharmacol Rev.* **67**, 821–870 (2015).
3. Tian, Y. *et al.* MicroRNA-27a promotes proliferation and suppresses apoptosis by targeting PLK2 in laryngeal carcinoma. *BMC Cancer* **14**,678 (2014).
4. Kim, J. *et al.* miR-27a and miR-27b regulate autophagic clearance of damaged mitochondria by targeting PTEN-induced putative kinase 1 (PINK1). *Mol Neurodegener.* **11**, 55 (2016).
5. Zhang, M., Sun, W., Zhou, M. & Tang, Y. MicroRNA-27a regulates hepatic lipid metabolism and alleviates NAFLD via repressing FAS and SCD1 *Sci Rep.* **7**, 14493 (2017)
6. Rahman, A., Sobia, R., Gupta, N., Kaer, L. & Das, N. Mycobacterium tuberculosis Subverts the TLR-2 - MyD88 Pathway to Facilitate Its Translocation into the Cytosol. *PLoS One.* **9**, e86886 (2014).
7. Ding, S., Wu, X., Li, G., Han, M., Zhuang, Y. & Xu, T. Efficient transposition of the piggyBac (PB) transposon in mammalian cells and mice. *Cell.* **122**, 473-483 (2005).
8. Ding, S., Xu, T. & Wu, X. Generation of genetically engineered mice by the piggyBac transposon system. *Methods Mol Biol.* **1194**,171-85 (2014).
9. Cui J, et al. Disruption of GPR45 causes reduced hypothalamic POMC expression and obesity. *J Clin Invest.* **126**, 3192-206 (2016).
10. Zhu X, Xie S, Xu T, Wu X, Han M. Rasal2 deficiency reduces adipogenesis and occurrence of obesity-related disorders. *Mol Metab.* **6**, 494-502 (2017).
11. Li G, Ye Z, Shi C, Sun L, Han M, Zhuang Y, Xu T, Zhao S & Wu X. The Histone Methyltransferase Ash11 is Required for Epidermal Homeostasis in Mice. *Sci Rep.* **7**, 45401 (2017).

12. Shi F, Ding S, Zhao S, Han M, Zhuang Y, Xu T, Wu X. A piggyBac insertion disrupts Foxl2 expression that mimics BPES syndrome in mice. *Hum Mol Genet.* **23**, 3792-800 (2014).
13. Ouimet, M. et al. Mycobacterium tuberculosis induces the miR-33 locus to reprogram autophagy and host lipid metabolism. *Nat. Immunol.* **17**, 677–686 (2016).
14. Berg, Jeremy; Tymoczko, John L.; Gatto, Gregory J.; Stryer, Lubert (2015). *Biochemistry* (Eighth ed.). New York, NY: W.H. Freeman and Company
15. Yang H et al., G protein-coupled receptor160 regulates mycobacteria entry into macrophages by activating ERK. *Cell Signal.* 2016 **28**, 1145-51 (2016).

Reviewers' Comments:

Reviewer #1:

Remarks to the Author:

The authors have adequately addressed the reviewer's concerns. Acceptance for publication is recommended.

Reviewer #2:

Remarks to the Author:

The manuscript by Liu et al remains novel and timeous. It has been improved by the current round of revisions. I now recommend publication.

1. Nature Communications uses a transparent peer review system, where for manuscripts submitted from January 2016 we are publishing the reviewer comments to the authors and author rebuttal letters of our research articles online as a supplementary peer review file. Please let us know in the cover letter when submitting the final version of your manuscript if you wish to opt out of this scheme or not. If you are concerned about the release of confidential data, we do permit redactions in the interest of confidentiality. If you would like to remove such information from these files, then please let us know specifically what information you would like to have removed. Please note that we cannot incorporate redactions for other reasons. Reviewer names will be published in the peer review files if the reviewer comments to the authors are signed by the reviewer, or if reviewers explicitly agree to release their name. For more information, please refer to our FAQ page at:

<https://media.nature.com/full/nature-assets/ncomms/authors/ncomms-transparent-peer-review.pdf>

We appreciate the transparent peer review system and would like to publish the peer review file. The response has been included in the cover letter.

2. Please ensure that an updated editorial policy checklist that verifies compliance with all required editorial policies is completed and uploaded with the revised article. All points on the policy checklist must be addressed; if needed, please revise your manuscript in response to these points. Please note that this form is a dynamic 'smart pdf' and must therefore be downloaded and completed in Adobe Reader.

Editorial policy checklist: <https://www.nature.com/authors/policies/Policy.pdf>

An updated editorial policy checklist have been completed and uploaded with the revised article. Meanwhile, all points on the policy checklist have been addressed and the manuscript has been revised accordingly when necessary.

3. Your manuscript should comply with our policies and format requirements, detailed in our checklist for authors at:

http://www.nature.com/article-assets/npg/ncomms/authors/ncomms_manuscript_checklist.pdf

The checklist have been completed and uploaded and the manuscript has been revised accordingly.

4. Please ensure that a Data Availability section at the end of the Methods section.

Data availability statements and data citations policy: All Nature Communications manuscripts must include a section titled "Data Availability"

as a separate section after the Methods section but before the References. For more information on this policy, and a list of examples, please see <http://www.nature.com/authors/policies/data/data-availability-statements-data-citations.pdf>

- Accession codes for deposited data
- Other unique identifiers (such as DOIs and hyperlinks for any other datasets)
- At a minimum, a statement confirming that all relevant data are available from the authors
- If applicable, a statement regarding data available with restrictions
- If a dataset has a Digital Object Identifier (DOI) as its unique identifier, we strongly encourage including this in the Reference list and citing the dataset in the Data Availability Statement.

*** DATA SOURCES:** We strongly encourage authors to deposit all new data associated with the paper in a persistent repository where they can be freely and enduringly accessed. We recommend submitting the data to discipline-specific, community-recognized repositories, where possible and a list of recommended repositories is provided here: <http://www.nature.com/sdata/policies/repositories>

If a community resource is unavailable, data can be submitted to generalist repositories such as figshare (<https://figshare.com/>) or Dryad Digital Repository (<http://datadryad.org/>). Please provide a unique identifier for the data (for example a DOI or a permanent URL) in the data availability statement, if possible. If the repository does not provide identifiers, we encourage authors to supply the search terms that will return the data. For data that have been obtained from publically available sources, please provide a URL and the specific data product name in the data availability statement. Data with a DOI should be further cited in the methods reference section.

Please refer to our data policies here:

<http://www.nature.com/authors/policies/availability.html>

Data availability statements have been included in the revised manuscript. In addition, the miRNA expression profiling data in our deep sequencing analysis are now available at Gene Expression Omnibus under accession code GSE119494, GSE119495, GSE119496.

5 To ensure correct hyperlinking of the accession codes in your manuscript, please add the hyperlink or DOI in square brackets directly after the code throughout (for example, '5XRN [<http://dx.doi.org/10.2210/pdb5XRN/pdb>]', '1483958 [<https://dx.doi.org/10.5517/ccdc.csd.cc1lt5m6>]', 'SRP109982 [<https://www.ncbi.nlm.nih.gov/sra/?term=SRP109982>]' or 'NQLW00000000 [https://www.ncbi.nlm.nih.gov/assembly/GCA_002312845.1/]').

We have corrected the hyperlinking of the accession code in the revised manuscript accordingly.

6 Please check whether your manuscript or Supplementary Information contain third-party images, such as figures from the literature, stock photos, clip art or commercial satellite and map data. We strongly discourage the use or adaptation of previously published images, but if this is unavoidable, please request the necessary rights documentation to re-use such material from the relevant copyright holders and return this to us when you submit your revised manuscript.

In particular, please indicate whether you or a co-author created figures

There is not any third-party image in our manuscript and supplementary information.

7. Nature journals require authors of life sciences research papers to include relevant details about several elements of experimental and analytical design in their manuscripts. This initiative aims to improve the transparency of reporting and the reproducibility of published results and is described at:

<http://www.nature.com/authors/policies/reporting.pdf> To ensure that your manuscript complies with our policy, please complete our checklist for authors: <https://www.nature.com/authors/policies/ReportingSummary.pdf>

You may also find the following collection of articles on statistics for biologists helpful: <http://www.nature.com/collections/qghhqm>

The reporting summary checklist has been completed and uploaded with the revised manuscript.

8. Please ensure that all affiliations are in the correct sequential order according to their position in the author list. Affiliation 1 must be associated with the first author. Please see this article for further detail:

<https://www.nature.com/articles/s41467-018-04254-0.pdf>

We have confirmed that all affiliations are in the correct sequential orders according to their position in the author list.

9. Please shorten the abstract to 150 words or fewer. It should be accessible and include the background and context of the work, ‘Here we report’ or an equivalent phrase, and then the major results and conclusions of the paper

written in the present tense. It must not contain references or unnecessary acronyms/abbreviations.

As per the suggestion, the abstract has been revised accordingly.

10. We do not permit a Materials and Methods section. Please rename to 'Methods'.

Following the suggestion, the “Materials and Methods” section has been renamed to “Methods”

11. The English language in your text would benefit from improvement for clarity and readability. We recommend that you either ask a colleague whose native language is English to review your manuscript or that you use one of the many English language editing services available. Two such services are provided by our affiliates Nature Research Editing Service (<http://bit.ly/NRES-NC>) and American Journal Experts (<http://bit.ly/AJE-NC>). Nature Communications authors are entitled to a 10% discount on their first submission to either of these services. To claim 10% off English editing from Nature Research Editing Service, follow this link (<http://bit.ly/NRES-10NC>). To claim 10% off American Journal Experts, follow this link (<http://bit.ly/AJE-10NC>).

As per the suggestion, we have improved the English language in our text with the assistance from native speakers.

12. Please remove, both in the abstract and the main text, phrases such as ‘new’, ‘novel’, ‘for the first time’, ‘unprecedented’, etc. as these are not needed to emphasise the importance of your work.

Following the suggestions, we have removed all the phrases such as ‘new’, ‘novel’, ‘for the first time’, ‘unprecedented’ and so on in the revised manuscript.

13. In the Methods, please provide sufficient information such that the experiments could reasonably be reproduced without reference to other papers, and avoid use of the term 'as described previously'.

As per the suggestion, we have included sufficient information in method and removed the term 'as described previously' in the revised manuscript.

13. Please include a complete list of all primers used in your study, including primer names and sequences, in either the Methods section or Supplementary Information.

All the primer names and sequences used in our study have been included in the Supplementary Table 1.

14. With regards to the experiments using human participants or data, please confirm that you have complied with all relevant ethical regulations and that a statement affirming this, and the name of the board and institution that approved the study protocol, is included in the methods section of the manuscript. Please also ensure that a statement is made in the methods confirming that informed consent was obtained from all human participants.

We have included all the information in the revised manuscript as requested.

15. With regards to the experiments using animal models, please confirm that you have complied with all relevant ethical regulations and that a statement affirming this, and the name of the board and institution that approved the study protocol, is included in the methods section of the manuscript.

We have included all the information in the revised manuscript as requested.

16. Please ensure that the origins of all cell lines used are stated in the methods section (ATCC, vendor, etc).

We have stated the origins of all cell lines used in our study in the methods section of the revised manuscript.

17. In the Methods, please ensure that the dilutions at which each antibody was used is stated, and catalogue numbers are provided for commercial antibodies.

We have provided that the information for the commercial antibodies including the dilutions and catalogue numbers in the methods section of the revised manuscript.

18. Please ensure that all blots and gels are accompanied by the locations of molecular weight/size markers and corresponding loading controls. Blots should be cropped such that at least one marker position is present. Please also supply uncropped scans of the most important blots as a supplementary figure in the Supplementary Information. This should be cited once in the Methods section.

We have confirmed all the blots and gels are accompanied by the locations of the molecular weight and all the uncropped scans of the blots are supplied as a supplementary figures.

19. In each Figure and Supplementary Figure where error bars are used, they must be defined, and the number of experimental replicates stated. One statement at the end of each figure is sufficient if the error bars are equivalent throughout the figure.

We have included all the required information in the revised manuscript.

20. Where p-values are presented as symbols/letters, please ensure that all symbols/letters are defined in the relevant figure legend, together with the statistical test used.

We have included all the required information in the revised manuscript.

21. Please ensure that at least one micrograph in each equivalent group in each figure is supplied with a representative scale bar, whose length is stated in the corresponding figure legend.

All of the micrographs have been supplied with representative scale bars and their length has been stated in the figure legend.

22. Please remove the scale bar labels from the figures in the main text and Supplementary Information (keeping the scale bar) and incorporate this information in the corresponding figure captions.

As per the suggestions, we have removed the scale bar labels from the figures in the whole manuscript and incorporate them in the figure legends.

23. Please ensure that all colour scales are defined in either the figure or its associated legend.

We have assured that all color scales have been defined.

24. Please ensure that the corresponding dot plots are overlaid in the bar charts.

We have confirmed that all the bar charts were overlaid with the dot plots.

25. Please ensure that the following issues are corrected:

-Figure 2: Panel “g” is referred to in the Fig. 2 legend, but there is no panel g in the figure.

As per the suggestion, we have deleted the “g” in the figure legend.

-Figure 3: Significance of red letters not defined in Fig. 3a.

As per the suggestion, we have correct the red letter in to black.

-Figure 4: Red circle and arrow not defined in Fig. 4c.

As per the suggestion, we have defined the red arrows in the figure legend accordingly.

-Scale bars in Figs. 1h, k, 4e, g, l and 6i should be defined in the legend and not on the actual figure itself.

As per the suggestion, we have reformatted the scale bars in these figures and defined them in the figure legends.

26. Please define any new abbreviations, symbols or colours present in your figures in the associated legends, noting that these should be written out in words (blue circles, red dashed line, etc.) as symbols will not appear properly in the HTML text.

We have assured that new abbreviations, symbols or colours present in the figures in the associated legends are clearly defined when necessary in the revised manuscript.

27. Please ensure that nonessential formatting is removed including highlighting.

We have confirmed that there is no nonessential formatting in the whole manuscript.

28. Please ensure the references are in the standard Nature format and follow the sequence: author list, title of paper, name of journal, volume number, initial-final page numbers or article number (year). Please note that dois are required only for online-only publications and correct journal abbreviations should be given.

We have reformatted the references according to the standard Nature format in the revised manuscript.

In particular:

-Please ensure that Ref. 25 (Kim et al. 2017) has its article number inserted.

As per the suggestions, the article number has been inserted in Ref. 25.

-Please provide a single continuous reference list (currently you provide a separate reference list 58 to 62 after Methods).

As per the suggestion, a single continuous reference list has been included.

*** Please supply an 'Author Contributions' section after the acknowledgement section that refers to all authors.**

The authors declare no competing interests. The 'Competing Interests' statement has been included in the revised manuscript.

29. Please make a 'Competing Interests' statement after the 'Author Contributions' section that refers to all authors. If there are no competing interests, please add the statement "The authors declare no competing interests."

The authors declare no competing interests. The 'Competing Interests' statement has been included in the revised manuscript.

30. Please note that we do not reformat Supplementary Information files; they will be uploaded with the published article as they are submitted with the final version of your manuscript. Please check everything very carefully and remove any track changes from the file. Failure to adhere to our style guidelines will result in delays in production. The only sections we permit in the Supplementary Information file are Supplementary Figures, Supplementary Tables, Supplementary Methods, Supplementary Notes, Supplementary Discussion, Supplementary References.

We have reformatted the Supplementary Information files adhering to the journal style guidelines.

31. Your paper will be accompanied by a two-sentence editor's summary, of between 250-300 characters, when it is published on our homepage. Could you please approve the draft summary below or provide us with a suitably edited version.

We thank for providing the draft summary and would like to edit it as following:

“How *Mycobacterium tuberculosis* (*Mtb*) escapes autophagy-mediated clearance is poorly understood. Here, Liu et al show that Mtb-induced *MicroRNA-27a* targets the ER-associated calcium transporter CACNA2D3, leading to suppression of antimicrobial autophagy and to enhanced intracellular survival of *Mtb*.”

32. As part of our efforts to communicate our content to a wider audience, we endeavour to highlight papers published in Nature Communications on the journal’s Twitter account (@NatureComms). If you would like us to mention authors, institutions or lab groups in these tweets, please provide the relevant twitter handles in your cover letter upon resubmission.

We appreciate the efforts in advertising the scientific findings. However, we don’t have any twitter handles to provide.